# Efficient Closed-Form Task Space Manipulability for a 7-DOF Serial Robot

**Gerold Huber \*** and **Dirk Wollherr**

Chair of Automatic Control Engineering (LSR), Department of Electrical and Computer Engineering, Technical University of Munich (TUM), 80333 Munich, Germany; dw@tum.de
* Correspondence: gerold.huber@tum.de

**Abstract:** With the increasing demand for robots to react and adapt to unforeseen events, it is essential that a robot preserves agility at all times. While *manipulability* is a common measure to quantify agility at a given joint configuration, an efficient direct evaluation in task space is usually not possible with conventional methods, especially for redundant robots with an infinite number of Inverse Kinematic solutions. Yet, this is essential for global online optimization of a robot posture. In this work, we derive analytical expressions for a conventional 7-degrees of freedom (7-DOF) serial robot structure, which enable the direct evaluation of manipulability from a reduced task space parametrization. The resulting expressions allow array operation and thus achieve very high computational efficiency with vector-optimized programming languages. This direct and simultaneous calculation of the *task space manipulability* for large numbers of poses benefits many optimization problems in robotic applications. We show applications in global optimization of robot mounting poses, as well as redundancy resolution with global online optimization w.r.t. manipulability.

**Keywords:** manipulability; inverse kinematics function; kinematic optimization; redundant robot; 7-DOF; redundancy resolution

## 1. Introduction

It is a common requirement in robotic manipulation tasks to quantify the capabilities of a robot at a given pose. Having such a scalar measure allows comparison of different kinematic configurations in terms of the chosen metric, and can be considered at a path planning as well as at a control level. While these measures are usually defined in terms of a given joint configuration [1–5], the task of the robot is typically not given in this *joint space*. For a general robot the *task space* is usually defined in SE(3), i.e., the space of 3D poses consisting of translation and rotation. For many practical problems it is thus relevant to directly evaluate this measure w.r.t. a parametrization of SE(3) rather than the joints. This requires combining the evaluation of the Inverse Kinematic (IK) with the selected capability metric. But direct calculation of the IK is always robot-dependent and general analytic solutions are not possible. This is especially true for redundant robots that have more degrees of freedom (DOF) in joint space than in task space and thus admit an infinite number of IK solutions for a given end-effector pose. While analytic IK solutions are well known for conventional 6-DOF kinematics [6], for general robotic structures numeric IK solvers are applied. However, they require several iterations to find an approximated joint configuration for a given end-effector pose. This is sufficient for calculating single poses, but it is inefficient for optimization problem solvers that require evaluation of large numbers of poses. This especially prevents time-critical computation of global optima. Expressions that can be evaluated directly are thus superior for fast computation. While an analytical IK for a general robot structure does not exist, our work focuses on the most commonly used articulated 6- and 7-DOF robot

serial kinematics. Yet, the 6 axis version can be viewed as a finite set of particular null space solutions of the 7-DOF.

### 1.1. Contribution

In this work, we develop a set of computationally efficient closed-form expressions to evaluate the *task space manipulability* of a 7-DOF serial robot structure, i.e., the mapping from a task space parametrization directly a manipulability measure. The main contributions of this work consist of:

1. a new parametrization of the state- and null space that results in concise IK expressions with symmetric structure in the individual components
2. analytical closed-form expressions from task space to manipulability measure w.r.t. joint limits, which allow *array operation* in vector-optimized programming languages. Note that array operation is also called *Vectorization* in e.g., *MATLAB*. It refers to the exploitation of Single Instruction Multiple Data (SIMD) instructions of modern Central Processing Unit (CPUs) and allows to operate on multiple data points simultaneously.
3. sensitivity analysis of manipulability in task space
4. real-time capable application for evaluating the task space manipulability of the entire null space, for globally optimal redundancy resolution w.r.t. manipulability of single poses and full trajectories on SE(3)

### 1.2. Related Work

For this concise review, we group previous work on the topic into the three areas: (1) performance measures in robotics, (2) direct methods for IK evaluation and algorithmic strategies on the velocity level, and (3) approaches for optimizing manipulability.

#### 1.2.1. Performance Measures

Arguably the most common performance measure for robot structures is the *manipulability* measure defined by Yoshikawa [1]. It is proportional to the volume of an ellipsoid, spanned by directional capabilities of a kinematic structure to generate velocities in task space at a given joint configuration. It is purely kinematic and does not consider any dynamic components. Yoshikawa also proposed a dynamic manipulability ellipsoid [2] on the acceleration level, for cases where dynamic effects cannot be neglected. This formulation was improved by Chiacchio et al. [3] to correctly account for gravity. A new formulation of a dynamic manipulability ellipsoid that better depicts the real manipulator capabilities in terms of task space accelerations was proposed by Chiacchio [4].

Besides manipulability on the velocity and acceleration levels due to mere kinematic relations, it is essential for practical applications to also consider joint limits as constraints directly on the position level. Vahrenkamp et al. [5] extended Yoshkawa's basic manipulability, by directly integrating joint limit penalization into the definition of the kinematic velocity Jacobian. This is achieved via a joint limit potential function.

Bong-Huan Jun et al. [7] introduce a task-oriented manipulability measure. While Yoshikawa's original measure [1] denotes the manipulability of the whole manipulator system, [7] considers manipulability w.r.t. to sub-tasks that only affect parts of the task space, e.g., axis specific tasks. Karim Abdel-Malek and Wei Yu [8] proposed an alternative dexterity measure for robot placement that does not depend on explicit IK solutions. They analyze an augmented Jacobian matrix that does not only hold information about position and orientation, but also joint limits of the end-effector. It represents the reachable workspace with surface patches and is computationally very demanding.

Our work has the aim of developing closed-form solutions that allow efficient array operation. For this reason, the task space manipulability formulation developed in this work applies Yoshikawa's original measure from [1]. Because its definition uses a determinant to map the joints to a scalar metric, it thus allows expansion to a continuous polynomial expression for efficient evaluation.

### 1.2.2. Inverse Kinematics

The IK problem of serial robot structures can be solved very elegantly on the velocity level, due to the linear relation of joint and task space velocities. However, numeric integration of the resulting joint velocities to joint angles needs stabilization against numerical drift and thus results in an iterative scheme. Originally proposed by Wolovich and Elliot [9], this group of IK solvers is nowadays typically referred to as Closed-Loop Inverse Kinematic (CLIK) solvers. Colomé and Torras [10] give an overview of the most common CLIK solvers, with an additional experimental comparison in terms of convergence, numerical error, singularity handling, joint limit avoidance, and the capability of reaching secondary goals. Antonelli [11] conducted a stability analysis of priority-based kinematic CLIK algorithms for redundant kinematics. He provides sufficient conditions for the control gains. While different stabilization schemes for CLIK solvers are proposed, the choice of gain parameters used in the control structure is rarely addressed. In practice these parameters are often empirically tuned. Bjoerlykhaug [12] proposes the use of a genetic algorithm for optimizing the feedback gain used in CLIK solvers, in order to minimize iteration cycles and maximize accuracy. In an experimental evaluation, he achieved a 50% decrease in computation time through his feedback gain tuning. Reiter et al. [13] propose a strategy for finding higher-order time-optimal IK solutions for redundant robots. They lay out solutions for fourth-order time derivatives of joint trajectories, applying a multiple shooting optimization method. This higher-order continuous differentiability is especially important for application on elastic mechanisms.

Siciliano [14] gives a tutorial on early common online IK algorithms. He states the important features of a direct inverse kinematics function, i.e., repeatability, cyclicity, or cyclic behaviour, and online applicability. Shimizu et al. [15] outline an analytical IK computation for a 7-DOF serial robot. The approach directly parametrizes the end-effector pose with Cartesian coordinates for translation and a rotation matrix for orientation. However, the use of the 2-quadrant atan function, as opposed to the 4-quadrant atan2 function, results in two problems. For one, the entire task space is not covered, and two, it results in discontinuous joint functions w.r.t. the null space parameter and thus leads to discontinuous IK solutions and corresponding null space limitations. A similar strategy, but extended to the entire domain, is proposed by Faria et al. [16]. They propose a position-based IK solution for a 7-DOF serial manipulator with joint limit and singularity avoidance.

Besides approaches that use kinematic insight of a structure, several machine learning algorithms are also considered in the literature. A detailed review is beyond the scope of this work, but we want to give a concise overview of research activities. D'Souza et al. [17] apply a locally weighted projection regression to learn the IK of a 30-DOF humanoid robot. This maps the non-convex problem onto a locally convex problem that is suitable for direct learning. Tejomurtula and Kak [18], as well as Köker et al. [19], applied artificial neural networks for finding an IK mapping for 3-DOF robots and showed the feasibility of the problem using conventional error-backpropagation and Kohonen networks. Sariyildiz et al. [20] compare support vector regression and artificial neural networks for learning IK mappings of a 7-DOF serial robot. They find that support vector regression is less prone to local minima and requires very few training data. Genetic algorithms were already early applied by Parker et al. [21]. They pointed out low positioning accuracy, but emphasize its simplicity in application. Köker [22] proposes a hybrid approach combining Elman neural networks with genetic algorithms. He was able to significantly improve accuracy for IK solutions of a 6-DOF mechanism in comparison to pure neural networks. Very recently, Dereli et al. [23] proposed a strategy to apply quantum behaved particle swarm optimization for finding IK solutions of a 7-DOF serial robot.

The IK expressions developed in this work are similar to the analytical approaches in [15,16] in terms of parametrizing the null space as arm angle. However, the new task space parametrization that we introduce results in more concise and, more importantly, fully vectorizable expressions that allow efficient array operations. In contrast to existing approaches in the literature, this computational advantage makes our approach suitable for simultaneous evaluation of a large number of poses.

### 1.2.3. Optimizing Manipulability

In conventional industrial contexts, optimizing cycling time is always of interest. Several publications deal with this problem, e.g., Kamrani et al. [24] use the Response Surface Method [25] to optimize robot placement w.r.t. cycling time. Chan and Dubey [26], as well as Dariush et al. [27], use a projection method of the joint limit gradient potential function. This is used for local manipulability optimization on the velocity level. Dufour and Suleiman [28] present an approach of integrating the manipulability index into an optimization-based IK solver, by using linear approximations of the nonlinear manipulability measure with numeric gradient calculations at every time step. Jin et al. [29] mention the difficulty of real-time manipulability optimization that is related to a high computational burden since the manipulability is a non-convex function to the joint angles of a robotic arm. Due to the capability of *high-speed parallel distributed processing*, they propose an approach using dynamic neural networks in order to implement manipulability optimization in real-time. Conducting computer simulations, they show that the proposed method raises the manipulability by almost 40% on average compared to existing methods.

Besides local optimization of a given joint configuration, for many robotic tasks it is required to include manipulability as criteria for optimization of the whole trajectory. Lee [30] shows that a required motion can be approximated by a series of manipulability ellipsoids. Guilamo et al. [31] present an algorithm for trajectory generation that maximizes the volume of the manipulability ellipsoid. Yoshikawa [1] already observed that the optimal postures of various manipulators form the viewpoint of manipulability, and often show resemblance of those naturally taken by human arms. This motivates the idea of manipulability transfer using a *learning by demonstration* strategy that is introduced by Rozo et al. [32]. Their approach allows robots to learn and reproduce a continuous set of manipulability ellipsoids by an expert's demonstration. In order to encode and retrieve those ellipsoids, they apply Gaussian Mixture Models and Gaussian Mixture Regression. In Jaquier et al. [33] the same authors exploit tensor-based representation, to consider that manipulability ellipsoids lie on the manifold of symmetric positive definite matrices. Faroni et al. [34] present an approach that maximizes the average manipulability of the overall task. Their method is based on the optimization of a cost function that depends on various points along a predetermined path. In particular, if the task of the manipulator is known a priori, this approach provides global manipulability optimization.

An approach for directly quantifying manipulability of a redundant robot in task space is proposed by Zacharias et al. [35]. They introduce a capability map, to guide the decision on how to place a mobile robot relative to an object. It is a sampling-based approach, based on the manipulability index. While the approach reveals in which regions the robot is capable of grasping objects from different angles, the information of optimal approaching directions is lost.

The task space manipulability approach in this work enables for the first time global manipulability optimization with real-time capabilities, due to its efficient formulation.

### 1.3. Outline

The remainder of the paper is organized as follows. The problem of a closed-loop task space manipulation framework is outlined in Section 2. In Section 3, the derivation of all analytical mappings is explained. Evaluation and analysis of the resulting task space manipulability is discussed in Section 4 and applied in global optimization formulations in Section 5. We conclude the work and outline future directions of development in Section 6.

## 2. Problem Formulation

Given a $n$-DOF serial robot, its forward kinematics

$$\text{FK}: \quad \mathbb{R}^n \to \text{SE}(3) \times \mathbb{R}^{n-6}, \quad \boldsymbol{q} \mapsto (\boldsymbol{z}, \boldsymbol{\lambda}) \tag{1}$$

maps the joints $\boldsymbol{q}$ onto the 3D end-effector pose $\boldsymbol{z}$ at a particular null space solution parametrized by $\boldsymbol{\lambda}$. To quantify the capability of moving in the SE(3) task space at a given joint configuration $\boldsymbol{q}$, a manipulability metric function

$$\text{M}: \quad \mathbb{R}^n \to \mathbb{R}^1, \quad \boldsymbol{q} \mapsto \mu \tag{2}$$

is applied. A proper choice of parametrization for $\boldsymbol{z}$ and $\boldsymbol{\lambda}$ assures the existence of the inverse function

$$\text{IK}: \text{SE}(3) \times \mathbb{R}^{n-6} \to \mathbb{R}^n, \quad (\boldsymbol{z}, \lambda) \mapsto \text{FK}^{-1}(\boldsymbol{z}, \lambda) =: \boldsymbol{q}. \tag{3}$$

We define the task space manipulability as the direct mapping

$$\text{M} \circ \text{IK}: \quad \text{SE}(3) \times \mathbb{R}^{n-6} \to \mathbb{R}^1, \quad (\boldsymbol{z}, \boldsymbol{\lambda}) \mapsto \mu \tag{4}$$

of a desired pose $\boldsymbol{z}$ in task space onto the manipulability measure $\mu$, considering all null space solutions parametrized by $\boldsymbol{\lambda}$. $(\text{M} \circ \text{IK})(\boldsymbol{z}, \boldsymbol{\lambda})$ denotes the function composition $\text{M}(\text{IK}(\boldsymbol{z}, \boldsymbol{\lambda}))$. Figure 1 illustrates the task space manipulability for a certain end-effector pose $\boldsymbol{z}$. Considering real-time critical online applications and feasibility of global optimization formulations, the development of the task space manipulability map can be broken down into three problems:

**Problem 1:** Find a parametrization of the task- and null space that exploits the kinematic structure for concise expressions.

**Problem 2:** Find closed-form expressions for all mappings from task space to manipulability that allow efficient array operation in vector-optimized programming languages.

**Problem 3:** Let $\mathcal{Q} \subset \mathbb{R}^7$ be the space of admissible joint configurations. Find an analytical expression of the range of the null space solutions $\Lambda(\boldsymbol{z}) := \{ \boldsymbol{\lambda} \in \mathbb{R}^{n-6} \mid \text{IK}(\boldsymbol{z}, \boldsymbol{\lambda}) \in (Q) \}$, for which the inverse kinematics function $\text{IK}(\cdot, \boldsymbol{\lambda})$ results in an admissible joint configuration $\boldsymbol{q} \in \mathcal{Q}$.

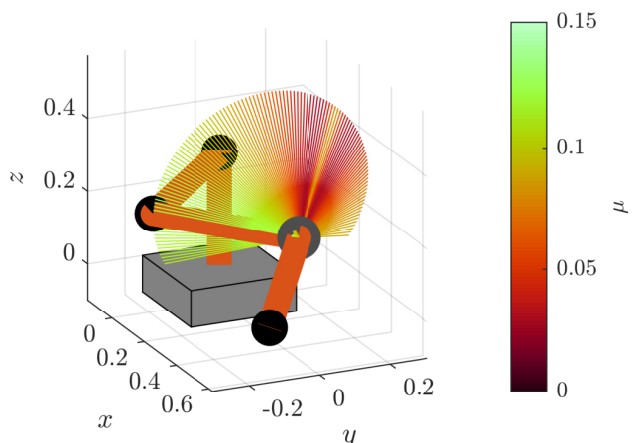

**Figure 1.** Illustration of the task space manipulability at a given end-effector pose. The null space of this 7-DOF S-R-S kinematics consists of the free elbow position (joint 4) along a circle. This position defines the direction of the forearm, i.e., the vector from the shoulder to the wrist. The colored fan shows all possible forearm poses with the corresponding manipulability color-coded from dark red (very bad) to light green (optimal). Colorless areas of the fan mark areas that violate joint constraints.

In this work, we investigate in detail the case of a 7-DOF serial robot kinematics in conventional Spherical-Revolute-Spherical (S-R-S) structure, such as the *KUKA LBR* series. In this context, S-R-S refers to a kinematic 7-DOF structure with alternating revolute joints, of which the rotation axes of the first and last 3 joints intersect. These two groups of intersecting axes behave kinematically like a spherical joint and are often referred to as shoulder and wrist. This type of kinematic structure leads to a 1-dimensional null space of solutions and thus $\lambda \in \mathbb{R}^1$.

## 3. Technical Approach

This section outlines the derivation of the closed-form task space manipulability for the considered special case of a 7-DOF serial robot kinematics. We first discuss the chosen manipulability mapping and possible reductions in joint space. Motivated by these reductions, we propose a task space projection onto a parameter space, which yields concise expressions for the IK. Figure 2 summarizes all developed mappings that are developed in this section. The section concludes with an analytic definition of the admissible null space at a given parameter end-effector pose.

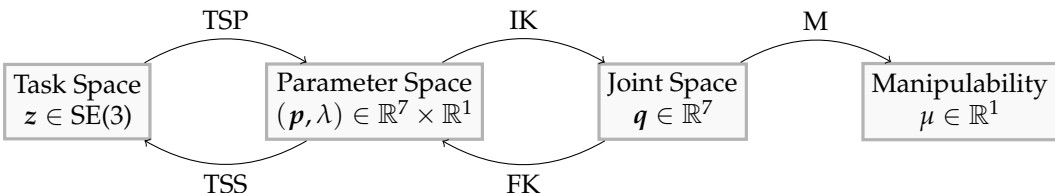

**Figure 2.** Relation of task space $z$, parameter space $p$, joint space $q$, and manipulability metric $\mu$. The mappings are referred to as Task Space Projection (TSP) and Task Space Surjection (TSS), Forward Kinematic (FK) and Inverse Kinematic (IK), and Manipulability (M).

Notational Notes

Scalars are written in plain lower case, vectors in bold-face lower case. Matrices are bold-face upper case, while plain upper case symbols refer to coordinate frames, mathematical spaces, and sets.

For vector indices, we use the common anthropomorphic analogy of a human arm. We refer to the origin of the kinematic as base $B$, and to positions of joint 2, joint 4, and joint 6 as shoulder $S$, elbow $E$, and wrist $W$ respectively. Body-fixed frames of the individual robot links are numbered 1 to 7 and relate to the bodies *after* the corresponding joints. The end-effector will be referred to as tool $T$.

Coordinate transformation matrices are written as $A_{kj}$ with 2 indices and are read from right to left, e.g., $A_{43}$ transforms the coordinate system from body-fixed frame of joint $q_3$ to joint $q_4$, whereas vector indices are read from left to right and their reference frame is written as left-hand side subscript. The notation $_B r_{SW}$ thus describes a vector $r$ pointing from shoulder $S$ to wrist $W$, expressed in base frame $B$. Cartesian base vectors of the coordinate systems are written as $\hat{x}$, $\hat{y}$, and $\hat{z}$. If a vector does not have a lower left index, it always refers to the base $B$.

### 3.1. Manipulability Measure

The central equation in robot kinematics is the linear forward velocity kinematic map

$$\dot{z}(q, \dot{q}) := J(q)\dot{q} \tag{5}$$

that relates general joint space velocities $\dot{q} \in \mathbb{R}^n$ to task space velocities $\dot{z} \in \mathbb{R}^6$, where the linear map $J(q) \in \mathbb{R}^{6 \times n}$ describes the velocity propagation from joint to task space at a given joint configuration $q \in \mathbb{R}^n$. It is defined by the kinematic chain and represents the derivative

$$J(q) = \frac{\partial \dot{z}(q, \dot{q})}{\partial \dot{q}}, \tag{6}$$

hence it is often referred to as *Robot Jacobian*.

Yoshikawa's manipulability measure [1], which we use in this work, is defined as

$$\mathrm{M}: \quad \mathbb{R}^n \to \mathbb{R}, \quad q \mapsto \sqrt{\det\left(J(q)J(q)^\top\right)} =: \mu \tag{7}$$

and is a measure, proportional to the volume of the velocity manipulability ellipsoid

$$\dot{q}^\top \dot{q} = 1 \tag{8a}$$

$$\dot{z}^\top (JJ^\top)^{-1} \dot{z} = 1. \tag{8b}$$

Note that (7) does not consider hardware-related joint limits. However, joint configurations that violate these constraints must not be considered.

Zlatanov et al. [36] explain that the forward velocity kinematic map (5) is not sufficient for exhaustive characterization of the singularities of a manipulator. Further, Staffetti et al. [37] show that many of these often-used manipulability indices are not invariant to change of reference frames, scale, or physical units. However, the big advantage of Yoshikawa's original manipulability metric is the fact that it can be expanded to a polynomial expression and thus qualifies for computationally efficient array operation. Further, derivatives can be calculated analytically. As outlined by Staffetti et al. [37], it is not a true metric for distance to a singularity but nonetheless serves as a relative comparison of manipulability qualities between joint configurations [38].

For a *n*-DOF serial robot kinematics, we refer to the $i = [1, n]$ absolute angular and translational velocities of the individual links, i.e., the velocity between the robot base $B$ and the body-fixed frame of link $i$, as $\omega_{Bi}$ and $v_{Bi}$. Expressed w.r.t. the link frame $i$, the velocities of the kinematic chain are calculated with

$$_i\omega_{Bi} = A_{ip}\,_p\omega_{Bp} + _i\omega_{pi} \tag{9a}$$

$$_iv_{Bi} = A_{ip}\left(_pv_{Bp} + _p\omega_{Bp} \times _pr_{pi}\right), \tag{9b}$$

where $p = i - 1$ is the predecessor link of $i$ and $(\times)$ denotes the cross product $\mathbb{R}^3 \times \mathbb{R}^3 \to \mathbb{R}^3$. In the following, *manipulability* refers to Yoshikawa's manipulability measure [1].

### 3.1.1. Reduction of First Joint

While Yoshikawa's manipulability measure is not invariant w.r.t. scale or physical units, it is in fact invariant to change of reference frames.

**Proof.** Given a vector of joint velocities $\dot{q}$ and task space velocities $\dot{z}$ w.r.t. to a reference frame $A$, the Jacobian matrix

$$_AJ(q) = \frac{\partial_A\dot{z}(q, \dot{q})}{\partial \dot{q}} \tag{10}$$

is used to define the manipulability index

$$_A\mu(q) = \sqrt{\det\left(_AJ(q)\,_AJ(q)^\top\right)}. \tag{11}$$

If this manipulability index is expressed in terms of a new reference frame $B$ via the block transformation matrix

$$A_{BA}^{\mathrm{blk}}(q) = \begin{bmatrix} A_{BA}(q) & 0 \\ 0 & A_{BA}(q) \end{bmatrix}, \tag{12}$$

consisting of rotation matrices $A_{BA}$, the manipulability index reads

$$_B\mu(q) = \sqrt{\det\left(A_{BA}^{\text{blk}}(q)\,_AJ(q)\left(A_{BA}^{\text{blk}}(q)\,_AJ(q)\right)^\top\right)}. \tag{13}$$

Considering the fact that Euclidean transformation matrices have $\det(A) = 1$, we find

$$_B\mu(q) = \sqrt{\det\left(_AJ(q)\,_AJ(q)^\top\right)} = \,_A\mu(q) \tag{14}$$

i.e., the manipulability measure $\mu$ is invariant to change of reference frames. $\quad\square$

If the reference frame is chosen to be fixed to any link after the first joint, it results in an expression for the manipulability measure that is independent of the first joint. This results from the fact that the first joint rotates the whole kinematic structure including the reference frame, but does not alter any geometric relations.

We consequently choose to formulate the Jacobian matrix w.r.t. to the end-effector frame, as this does not only lead to the independence of $q_1$, but also results in the most concise expression.

### 3.1.2. Reduction of Last Joint

For a special case of a 7-DOF serial kinematic, the parameter space of the manipulability can be further reduced. This special case consists of kinematic structures, whose origin of the end-effector frame lies on the rotation axis of the last joint $q_n$. The purely angular contribution of $q_n$ does not alter the kinematic configuration but only rotates the reference frame and with it the manipulability ellipsoid. The shape of the ellipsoid is not affected and so $q_n$ can also not influence the manipulability measure.

### 3.1.3. Closed-Form Expression

Exploiting these two reductions by formulating the $_TJ$ w.r.t. to the end-effector frame $T$ and assuming the tool center point (TCP) along the last joint axis, it is possible to expand the entire determinant expression of the matrix $_TJ\,_TJ^\top \in \mathbb{R}^{6\times6}$ from (7) to a symbolic polynomial expression using, e.g., *MATLAB Symbolic Math Toolbox*™. The advantage being that, unlike the original matrix expression, the polynomial form allows array operation in vector-optimized programming languages. This enables simultaneous evaluation of an entire set of joint configurations. The full manipulability function is listed in Appendix A.

### *3.2. Task Space Parametrization*

The decision of choosing a parametrization for the SE(3) pose, as well as the 1D null space, is essential for the derivation of concise analytical formulations. We propose the following parameter requirements (PR) for a suitable parametrization in regard to the IK functions. The parameter set must

PR1: uniquely define the null space parameter for the entire space of SE(3).
PR2: result in a minimal number of parameters for the components of the IK vector map $p \mapsto q$.
PR3: allow direct application of the above-mentioned reductions.

Different approaches for null space parametrization were proposed in the literature. The redundancy is either directly parametrized by a redundant joint [39,40], or more commonly by a joint-independent arm angle [15,41]. Shimizu et al. [15] argued that joint-based parametrization is not suitable for the discussed 7-DOF S-R-S mechanism due to possible ambiguous results. Kreutz-Delgado et al. [41] define the arm angle as the angle between an arm and a reference plane. The arm plane is spanned by shoulder, elbow, and wrist locations. The reference plane is defined by a

fixed vector and the vector from shoulder to wrist. Shimizu et al. [15] point out arithmetic singularities in the original definition whenever the two vectors are collinear. They enhance the robustness of the definition by defining the reference plane in terms of a particular solution $q_3 \overset{!}{=} 0$, which resembles the solution of conventional non-redundant 6-DOF mechanisms. While this definition is unique w.r.t. the conservative joint limits of their analyzed robot structure, it is ambiguous whenever the reduced non-redundant 6-DOF mechanism admits multiple configurations that result in the same end-effector pose.

In this work, we introduce a parametrization that fulfills all the above-discussed parameter requirements. Figure 3 illustrates the following discussion. Independent of a desired end-effector pose, positions of the base $B$ and shoulder $S$ are always stationary, where

$$_B\boldsymbol{r}_{BS} := (l_B + l_1)\hat{\boldsymbol{z}} \tag{15}$$

with link lengths of the base link $l_B$ and the first link $l_1$. Additionally, defining a desired end-effector pose relative to the robot base in SE(3), consisting of $_B\boldsymbol{r}_{BT}$ for translation and $\boldsymbol{A}_{TB}$ for orientation, determines not only the location of the tool-center-point $T$ but also the wrist position

$$_B\boldsymbol{r}_{BW} := {}_B\boldsymbol{r}_{BT} - \boldsymbol{A}_{B6}(l_6 + l_7 + l_T)\hat{\boldsymbol{z}}, \tag{16}$$

with link lengths $l_6$ and $l_7$, and a potential tool length $l_T$. This wrist position is used for define the translational component of the end-effector pose $\boldsymbol{z}$. The position $_B\boldsymbol{r}_{SW}$ is parametrized by spherical coordinates $(r_{\mathrm{ref}}, \gamma_{\mathrm{ref}}, \beta_{\mathrm{ref}})$ with coordinate plane $_B\hat{\boldsymbol{x}}\hat{\boldsymbol{z}}$, origin $S$ and $_B\hat{\boldsymbol{z}}$ as polar axis. The parameters are radius $r_{\mathrm{ref}}$, longitudinal angle $\gamma_{\mathrm{ref}}$, and azimuthal angle $\beta_{\mathrm{ref}}$. Note that $\gamma_{\mathrm{ref}}$ and $\beta_{\mathrm{ref}}$ directly align with the rotation axis of $q_1$ and $q_2$. These two angles also define the reference frame $R$ with

$$\boldsymbol{A}_{RB}(\gamma_{\mathrm{ref}}, \beta_{\mathrm{ref}}) := \boldsymbol{A}_y(\beta_{\mathrm{ref}})\boldsymbol{A}_z(\gamma_{\mathrm{ref}}). \tag{17}$$

The orientation is parametrized along a consecutive Euler angle sequence $Z \to Y' \to Z''$, which again corresponds to the sequence of the joint structure. However, instead of directly parametrizing $\boldsymbol{A}_{TB}$, we parametrize the end-effector orientation with respect to the reference frame, i.e.,

$$\boldsymbol{A}_{TR}(\gamma_{\mathrm{EE}}, \beta_{\mathrm{EE}}, \psi_{\mathrm{EE}}) := \boldsymbol{A}_z(\psi_{\mathrm{EE}})\boldsymbol{A}_y(\beta_{\mathrm{EE}})\boldsymbol{A}_z(\gamma_{\mathrm{EE}}). \tag{18}$$

Regarding the stated parameter requirement PR2, this makes the IK functions of the wrist angles $(q_5, q_6, q_7)$ as independent of the shoulder parameters $(r_{\mathrm{ref}}, \gamma_{\mathrm{ref}}, \beta_{\mathrm{ref}})$ as possible, as will be seen in the IK Section 3.3.

The 1D null space is parametrized by the arm angle $\lambda$. In contrst to Shimizu et al. [15] we do not define the arm angle w.r.t. to the non-redundant solution $q_3 \overset{!}{=} 0$, but w.r.t. to the introduced reference frame $R$. Let $\lambda$ be the arm angle, which defines a new frame $L$ with

$$\boldsymbol{A}_{LB}(\gamma_{\mathrm{ref}}, \beta_{\mathrm{ref}}, \lambda) := \boldsymbol{A}_z(\lambda)\boldsymbol{A}_{RB}(\gamma_{\mathrm{ref}}, \beta_{\mathrm{ref}}), \tag{19}$$

such that the negative frame base vector $(-_L\hat{\boldsymbol{x}})$ points in direction of the elbow $E$. This uniquely defines the null space parameter as required in PR1. The full set of parameters is thus given with tuple $(\boldsymbol{p}, \lambda) \in \mathbb{R}^6 \times \mathbb{R}$, consisting of the parameter vector

$$\boldsymbol{p} := [r_{\mathrm{ref}}, \gamma_{\mathrm{ref}}, \beta_{\mathrm{ref}}, \gamma_{\mathrm{EE}}, \beta_{\mathrm{EE}}, \psi_{\mathrm{EE}}]^\top \tag{20}$$

and arm angle $\lambda$. The individual parameter range definitions are

$$
\begin{aligned}
r_{\text{ref}} &\in [\quad r_{\text{ref}}^{\min}, \quad r_{\text{ref}}^{\max} \quad] \\
\gamma_{\text{ref}} &\in [\quad -\pi, \quad +\pi \quad] \\
\beta_{\text{ref}} &\in [\quad 0, \quad +\pi \quad] \\
\gamma_{\text{EE}} &\in [\quad -\pi, \quad +\pi \quad] \\
\beta_{\text{EE}} &\in [\quad 0, \quad +\pi \quad] \\
\psi_{\text{EE}} &\in [\quad -\pi, \quad +\pi \quad] \\
\lambda &\in [\quad -\pi, \quad +\pi \quad]
\end{aligned}
\tag{21}
$$

and form the parameter space $\mathcal{P} \subset \mathbb{R}^7$. Note that the two parameters $\gamma_{\text{ref}}$ and $\psi_{\text{EE}}$ solely affect joints $q_1$ and $q_7$, which do not influence manipulability. The task space manipulability developed in this work can thus without loss of information be represented by the reduced parameter vector $\boldsymbol{p}^{\text{red}} \in \mathcal{P}^{\text{red}} \subset \mathbb{R}^4$ consisting of

$$
\boldsymbol{p}^{\text{red}} := [r_{\text{ref}}, \beta_{\text{ref}}, \gamma_{\text{EE}}, \beta_{\text{EE}}]^\top .
\tag{22}
$$

This complies with the stated requirement PR3. The presented parametrization is the fundamental core for the concise mappings developed in the remaining section.

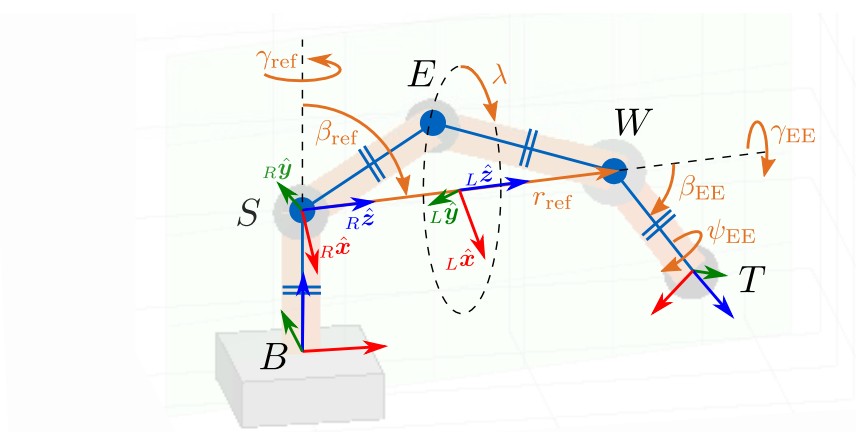

**Figure 3.** Parametrization of the Task Space. Positions of Base $B$ and Shoulder $S$ are fixed. Translation reference parameters $(r_{\text{ref}}, \gamma_{\text{ref}}, \beta_{\text{ref}})$ define the position of the Wrist $W$. The end-effector parameters $(\gamma_{\text{EE}}, \beta_{\text{EE}}, \psi_{\text{EE}})$ describe the rotation from reference frame $R$ to tool frame $T$ as consecutive $Z \rightarrow Y' \rightarrow Z''$ Euler angles. The null space is parametrized with $\lambda$. It defines the position of the elbow $E$ via relative rotation between the elbow oriented frame $L$ and frame $R$.

### 3.2.1. Task Space Projection

We refer to the extraction of the parameter vector $\boldsymbol{p} = [r_{\text{ref}}, \gamma_{\text{ref}}, \beta_{\text{ref}}, \gamma_{\text{EE}}, \beta_{\text{EE}}, \psi_{\text{EE}}]^\top$ from a given end-effector pose $\boldsymbol{z} \in \text{SE}(3)$ as *Task Space Projection*. Without loss of generality, we assume the pose $\boldsymbol{z} \in \text{SE}(3)$ is described with Cartesian Coordinates $(x, y, z)$ for translation ${}_B\boldsymbol{r}_{BT}$ together with a Rotation matrix $\boldsymbol{A}_{TB}$ for orientation. As a reference matrix for extracting the parameter space angles $(\gamma_{\text{EE}}, \beta_{\text{EE}}, \psi_{\text{EE}})$ we state the rotation matrix for a general $ZYZ$ Euler sequence

$$
\boldsymbol{A}_{zyz}(\gamma, \beta, \psi) := \boldsymbol{A}_z(\psi)\boldsymbol{A}_y(\beta)\boldsymbol{A}_z(\gamma) =
$$
$$
\begin{pmatrix}
\text{c}(\beta)\,\text{c}(\gamma)\,\text{c}(\psi) - \text{s}(\gamma)\,\text{s}(\psi) & \text{c}(\gamma)\,\text{s}(\psi) + \text{c}(\beta)\,\text{c}(\psi)\,\text{s}(\gamma) & -\text{c}(\psi)\,\text{s}(\beta) \\
-\text{c}(\psi)\,\text{s}(\gamma) - \text{c}(\beta)\,\text{c}(\gamma)\,\text{s}(\psi) & \text{c}(\gamma)\,\text{c}(\psi) - \text{c}(\beta)\,\text{s}(\gamma)\,\text{s}(\psi) & \text{s}(\beta)\,\text{s}(\psi) \\
\text{c}(\gamma)\,\text{s}(\beta) & \text{s}(\beta)\,\text{s}(\gamma) & \text{c}(\beta)
\end{pmatrix}
\tag{23}
$$

that shows that we can define a mapping $\mathrm{eul}_{ZYZ} : \mathrm{SE}(3) \to \mathbb{R}^3$ as

$$
\mathrm{eul}_{ZYZ} : \quad \mathrm{SE}(3) \to \mathbb{R}^3, \quad A_{zyz} \mapsto
\begin{bmatrix}
\mathrm{atan2}\left( \left[ A_{zyz}(z) \right]_{(3,2)}, \left[ A_{zyz}(z) \right]_{(3,1)} \right) \\
\arccos \left( \left[ A_{zyz}(z) \right]_{(3,3)} \right) \\
\mathrm{atan2}\left( \left[ A_{zyz}(z) \right]_{(2,3)}, - \left[ A_{zyz}(z) \right]_{(1,3)} \right)
\end{bmatrix}
=:
\begin{bmatrix} \gamma \\ \beta \\ \psi \end{bmatrix}
\tag{24}
$$

that extracts the Euler angles from a rotation matrix in SE(3). The operator $[ \cdot ]_{(i,j)}$ returns the element at row $i$ and column $j$ of a matrix.

The Task Space Projection

$$
\mathrm{TSP} : \quad \mathrm{SE}(3) \to \mathbb{R}^6, \quad z \mapsto p
\tag{25a}
$$

consists of the mappings

$$
r_{\mathrm{ref}}(z) := \| {}_B r_{SW} \|_2
\tag{25b}
$$

$$
\beta_{\mathrm{ref}}(z) := \frac{\pi}{2} - \arctan \frac{\left[ {}_B r_{SW} \right]_{(3)}}{\left[ {}_B r_{SW} \right]_{(1)}}
\tag{25c}
$$

$$
\gamma_{\mathrm{ref}}(z) := \mathrm{atan2}\left( \left[ {}_B r_{SW} \right]_{(2)}, \left[ {}_B r_{SW} \right]_{(1)} \right)
\tag{25d}
$$

$$
\begin{bmatrix} \gamma_{\mathrm{EE}} \\ \beta_{\mathrm{EE}} \\ \psi_{\mathrm{EE}} \end{bmatrix} (z) := \mathrm{eul}_{ZYZ}( A_{7R}(z, \gamma_{\mathrm{ref}}, \beta_{\mathrm{ref}})).
\tag{25e}
$$

With the shoulder-wrist vector

$$
\begin{aligned}
{}_B r_{SW} &:= {}_B r_{BR} - {}_B r_{BS} \\
&= {}_B r_{BT} - A_{B6}\, {}_6\hat{z}(l_6 + l_7 + l_T) - {}_B\hat{z}(l_B + l_1).
\end{aligned}
\tag{26}
$$

and the rotation matrix

$$
A_{7R}(z, \gamma_{\mathrm{ref}}, \beta_{\mathrm{ref}}) := A_{7T} A_{TB}(z) A_{BR}(\gamma_{\mathrm{ref}}, \beta_{\mathrm{ref}}),
\tag{27}
$$

derived from the desired task space pose. Rotation $A_{7T}$ is the constant rotation matrix from body fixed frame of link 7 to the TCP frame.

### 3.2.2. Task Space Surjection

We refer to the inverse mapping, i.e., from the parameter vector $p$ to the task space pose $z$, as Task Space Surjection (TSS)

$$
\mathrm{TSS} : \quad \mathbb{R}^6 \to \mathrm{SE}(3), \quad p \mapsto z.
\tag{28a}
$$

The relations are given with

$$
{}_B r_{BT} := (l_B + l_1)\, {}_B\hat{z} + A_{BR}(\gamma_{\mathrm{ref}}, \beta_{\mathrm{ref}}) \left( {}_R\hat{z} r_{\mathrm{ref}} + A_{R7}(\gamma_{\mathrm{EE}}, \beta_{\mathrm{EE}}, \psi_{\mathrm{EE}})(l_6 + l_7 + A_{7T} l_T) \right)
\tag{28b}
$$

$$
A_{TB} := A_{7T} A_{7R}(\gamma_{\mathrm{EE}}, \beta_{\mathrm{EE}}, \psi_{\mathrm{EE}}) A_{RB}(\gamma_{\mathrm{ref}}, \beta_{\mathrm{ref}})
\tag{28c}
$$

using the established definitions in the previous sections.

### 3.3. Inverse Kinematics

In this section we derive closed-form expressions for the IK map. After discussing the choice of the default manipulator configuration, we derive the individual IK mappings of the robot joints. Corresponding to the S-R-S structure, we group the joints into shoulder angles $\{q_1, q_2, q_3\}$, the elbow angle $\{q_4\}$, and wrist angles $\{q_5, q_6, q_7\}$.

### 3.3.1. Manipulator Configuration

Due to the possible reconfiguration of the robot kinematics, i.e., whenever 3 revolute joint axes intersect in one point, with 2 being coaxial and the third being perpendicular to the links, there exists an alternative configuration

$$\text{FK}(\text{coaxial}_1, \text{perpendicular}, \text{coaxial}_2) = \text{FK}(\text{coaxial}_1 + \pi, -\text{perpendicular}_2, \text{coaxial}_2 + \pi) \quad (29)$$

that results in the same FK. In the 7-DOF S-R-S structure considered in this work, this is the case for the tuples $(q_1, q_2, q_3)$, $(q_3, q_4, q_5)$, and $(q_5, q_6, q_7)$. Therefore, defining only the end-effector pose as well as the elbow position results in 8 possible configurations. Of course, it is important to derive an IK map that results in one specific configuration for the entire parameter space. The following derivation is designed to yield in a configuration as depicted in Figure 3 for the default case $q_1 = q_3 = q_5 = 0$. This is achieved by choosing the joint angle ranges

$$\begin{aligned}
q_1 &\in [\quad -\pi, \quad +\pi \quad] \\
q_2 &\in [\quad\quad 0, \quad +\pi \quad] \\
q_3 &\in [\quad -\pi, \quad +\pi \quad] \\
q_4 &\in [\quad\quad 0, \quad +\pi \quad] \\
q_5 &\in [\quad -\pi, \quad +\pi \quad] \\
q_6 &\in [\quad\quad 0, \quad +\pi \quad] \\
q_7 &\in [\quad -\pi, \quad +\pi \quad].
\end{aligned} \quad (30)$$

We refer to this definition as $\mathcal{Q}_{\text{sc}} \subset \mathbb{R}^7$, i.e., the space of joints in standard configuration.

### 3.3.2. Elbow Angles

The central geometric shape to express the arm portion of joints is the triangle $\overline{SEW}$ as depicted in Figure 3. It is fully defined by the parameter $r_{\text{ref}}$, as well as the robot related constant link lengths

$$r_{SE} := l_3 + l_4 \quad (31)$$

$$r_{EW} := l_5 + l_6. \quad (32)$$

The law of cosines in this triangle allows direct calculation of joint 4

$$r_{\text{ref}}^2 = r_{SE}^2 + r_{EW}^2 - r_{SE} r_{EW} \cos(\pi - q_4) \quad (33)$$

$$q_4(r_{\text{ref}}) := \pi - \arccos\left(\frac{r_{SE}^2 + r_{EW}^2 - r_{\text{ref}}^2}{2 r_{SE} r_{EW}}\right) \quad (34)$$

as well as the adjoint angles

$$r_{SE} = r_{\text{ref}}^2 + r_{EW}^2 - r_{\text{ref}} r_{EW} \cos(\theta_S) \quad (35)$$

$$\theta_S(r_{\text{ref}}) := \arccos\left(\frac{r_{\text{ref}}^2 + r_{EW}^2 - r_{SE}}{2 r_{\text{ref}} r_{EW}}\right) \quad (36)$$

and

$$r_{EW} = r_{\text{ref}}^2 + r_{SE}^2 - r_{\text{ref}} r_{SE} \cos(\theta_W) \tag{37}$$

$$\theta_W(r_{\text{ref}}) := \arccos\left(\frac{r_{\text{ref}}^2 + r_{SE}^2 - r_{EW}}{2 r_{\text{ref}} r_{SE}}\right). \tag{38}$$

The latter are used to define alternative rotation frame compositions for the derivation of the remaining joints. See Figure 4 for an overview of the relations between all introduced coordinate frames.

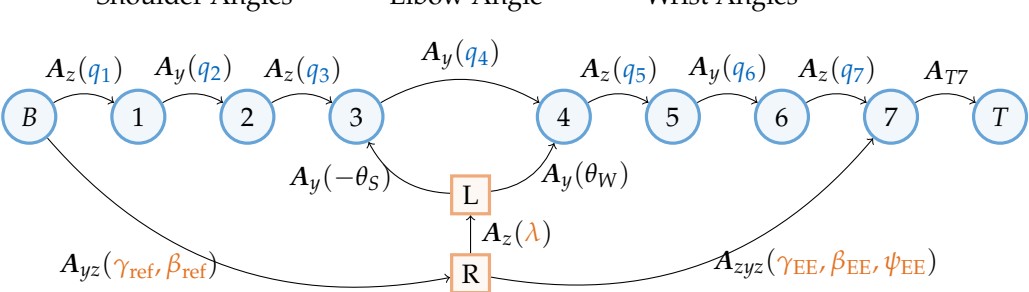

**Figure 4.** Reference frames and their relations. The blue frames *B* to *T* are fixed to the corresponding body-fixed coordinate systems of the robot links. Orange frames *R* and *L* are additional reference frames for the introduced parameter space. The arrows mark the rotations between the frames of reference.

### 3.3.3. Shoulder Angles

Reusing the $ZYZ$ Euler sequence extraction function (24) makes it possible to directly define the IK function of the shoulder angles $\{q1, q2, q3\}$. The parameter-related frames *R* and *L* (cf. Figure 4) are used to compose the transformation matrix

$$\boldsymbol{A}_{3B}(\boldsymbol{p}, \lambda) := \boldsymbol{A}_y(-\theta_W) \boldsymbol{A}_z(\lambda) \boldsymbol{A}_{RB}(\beta_{\text{ref}}, \gamma_{\text{ref}}) \tag{39}$$

and extract

$$\begin{bmatrix} q_1 \\ q_2 \\ q_3 \end{bmatrix}(\boldsymbol{p}, \lambda) := \text{eul}_{ZYZ}(\boldsymbol{A}_{3B}(\boldsymbol{p}, \lambda)). \tag{40}$$

### 3.3.4. Wrist Angles

Analogously to the shoulder angles, the wrist angles $\{q_5, q_6, q_7\}$ can be calculated by composing the transformation matrix

$$\boldsymbol{A}_{74}(\boldsymbol{p}, \lambda) := \boldsymbol{A}_{7R}(\gamma_{\text{EE}}, \beta_{\text{EE}}, \psi_{\text{EE}}) \boldsymbol{A}_z(-\lambda) \boldsymbol{A}_y(-\theta_S) \tag{41}$$

and extracting the wrist angles with

$$\begin{bmatrix} q_5 \\ q_6 \\ q_7 \end{bmatrix}(\boldsymbol{p}, \lambda) := \text{eul}_{ZYZ}(\boldsymbol{A}_{74}(\boldsymbol{p})). \tag{42}$$

### 3.3.5. Overview

All closed-form expressions resulting from the IK mapping are fully listed in Appendix B. The parameter dependencies of the individual function components are

$$\text{IK}_1: \quad \mathbb{R}^4 \to \mathbb{R}^1, \qquad (\theta_S(r_{\text{ref}}), \gamma_{\text{ref}}, \beta_{\text{ref}}, \lambda) \mapsto q_1 \tag{43a}$$

$$\text{IK}_2: \quad \mathbb{R}^3 \to \mathbb{R}^1, \qquad (\theta_S(r_{\text{ref}}), \beta_{\text{ref}}, \lambda) \mapsto q_2 \tag{43b}$$

$$\text{IK}_3: \quad \mathbb{R}^3 \to \mathbb{R}^1, \qquad (\theta_S(r_{\text{ref}}), \beta_{\text{ref}}, \lambda) \mapsto q_3 \tag{43c}$$

$$\text{IK}_4: \quad \mathbb{R}^1 \to \mathbb{R}^1, \qquad\qquad (r_{\text{ref}}) \mapsto q_4 \tag{43d}$$

$$\text{IK}_5: \quad \mathbb{R}^3 \to \mathbb{R}^1, \qquad (\theta_W(r_{\text{ref}}), \gamma_{\text{EE}}, \beta_{\text{EE}} - \lambda) \mapsto q_5 \tag{43e}$$

$$\text{IK}_6: \quad \mathbb{R}^3 \to \mathbb{R}^1, \qquad (\theta_W(r_{\text{ref}}), \gamma_{\text{EE}}, \beta_{\text{EE}} - \lambda) \mapsto q_6 \tag{43f}$$

$$\text{IK}_7: \quad \mathbb{R}^4 \to \mathbb{R}^1, \quad (\theta_W(r_{\text{ref}}), \gamma_{\text{EE}}, \beta_{\text{EE}}, \psi_{\text{EE}} - \lambda) \mapsto q_7 \tag{43g}$$

and show the low dimensional dependency as required by PR2. Note that parameters $\gamma_{\text{ref}}$ and $\psi_{\text{EE}}$ do solely influence $q_1$ and $q_7$ resp., and thus do not influence manipulability. Further, in this formulation the shoulder and wrist joints result in equivalent mappings, with symmetrical assignments. Their relations are given as

$$\text{IK}_5 = \text{IK}_3(\theta_W(r_{\text{ref}}), \beta_{\text{EE}}, \gamma_{\text{EE}} - \lambda) \tag{44a}$$

$$\text{IK}_6 = \text{IK}_2(\theta_W(r_{\text{ref}}), \beta_{\text{EE}}, \gamma_{\text{EE}} - \lambda) \tag{44b}$$

$$\text{IK}_7 = \text{IK}_1(\theta_W(r_{\text{ref}}), \psi_{\text{EE}}, \beta_{\text{EE}}, \gamma_{\text{EE}} - \lambda). \tag{44c}$$

This is an interesting geometrical insight that results from the chosen parameter set.

### 3.4. Forward Kinematics

Although not used in the task space manipulability mapping, we state—for the sake of completeness—the forward mapping

$$\text{FK}: \quad \mathbb{R}^7 \to \mathbb{R}^6 \times \mathbb{R}, \quad q \mapsto (p, \lambda) \tag{45}$$

using the developed relations from the previous section on the IK problem. From the elbow angle $q_4$ and the relation (33), $r_{\text{ref}}$ is mapped by

$$r_{\text{ref}}(q_4) := \sqrt{r_{SE}^2 + r_{EW}^2 - r_{SE} r_{EW} \cos(\pi - q_4)}. \tag{46}$$

The Euler angle extraction function (24) allows again a concise definition of the remaining mappings. The shoulder joints $\{q_1, q_2, q_3\}$ with the adjoint shoulder angle $\theta_S(r_{\text{ref}})$ from (35) parametrize

$$\begin{bmatrix} \gamma_{\text{ref}} \\ \beta_{\text{ref}} \\ \lambda \end{bmatrix} := \text{eul}_{ZYZ}\left(A_{LB}(q, r_{\text{ref}})\right) \tag{47a}$$

where

$$A_{LB}(q, r_{\text{ref}}) := A_y(\theta_S(r_{\text{ref}})) A_z(q_3) A_y(q_2) A_z(q_1). \tag{47b}$$

Analogously, the wrist joints $\{q_5, q_6, q_7\}$ and the adjoint wrist angle $\theta_W(r_{\text{ref}})$ from (37) define the end-effector parameters

$$\begin{bmatrix} \lambda + \gamma_{\text{EE}} \\ \beta_{\text{EE}} \\ \psi_{\text{EE}} \end{bmatrix} := \text{eul}_{ZYZ} \left( \boldsymbol{A}_{7L}(\boldsymbol{q}, r_{\text{ref}}) \right) \tag{48a}$$

where

$$\boldsymbol{A}_{7L}(\boldsymbol{q}, r_{\text{ref}}) := \boldsymbol{A}_z(q_7) \boldsymbol{A}_y(q_6) \boldsymbol{A}_z(q_5) \boldsymbol{A}_y(-\theta_W(r_{\text{ref}})). \tag{48b}$$

The composition of rotations is in accordance with the structural relation depicted in Figure 4. This concludes the FK problem.

*3.5. Admissible Parameter Space*

The compact analytical expressions also allow solving analytically for an upper and lower bound of $\lambda$, given maximal joint angles $q_i^{\max}$. Let $\mathcal{Q} := \left\{ \boldsymbol{q} \mid \boldsymbol{q} \in \mathcal{Q}_{\text{sc}}, \ |q_i| \leq q_i^{\max} \right\}$ be the space of admissible joint configurations. In this section, we determine the space of admissible parameters

$$\mathcal{A} := \{ (\boldsymbol{p}, \lambda) \mid \text{IK}(\boldsymbol{p}, \lambda) \in \mathcal{Q} \}. \tag{49}$$

Recall the definition of the parameter vector $\boldsymbol{p} := [r_{\text{ref}}, \gamma_{\text{ref}}, \beta_{\text{ref}}, \gamma_{\text{EE}}, \beta_{\text{EE}}, \psi_{\text{EE}}]^\top$ from (20). Only $r_{\text{ref}}$ is of linear nature and thus has a limited range. The remaining parameters describe angles and hence need not be limited. While $\text{IK}_4$ directly relates joint limits of the elbow joint with the admissible range of $r_{\text{ref}}$, the null space parameter $\lambda$ is related to all remaining joints. Each of which can potentially exclude partitions of the full range of $\lambda$. The set of admissible parameters $\mathcal{A}$ must consider all joint limits and results from the intersection

$$\mathcal{A} = \bigcap_{i=1}^n \mathcal{A}_i, \tag{50}$$

of the $n$ individual joint-related portions.

3.5.1. Shoulder-Wrist Distance $r_{\text{ref}}$

Elbow joint 4 directly limits the parameter $r_{\text{ref}}$. Solving (43d) for $r_{\text{ref}}$ gives

$$r_{\text{ref}}(q_4) := \sqrt{r_{SE}^2 + r_{EW}^2 - 2r_{SE}r_{EW}\cos\left(\pi - q_4\right)} \tag{51}$$

and defines the lower and upper bounds

$$r_{\text{ref}}(q_4^{\max}) \ \leq r_{\text{ref}} \leq \ r_{\text{ref}}(0) \tag{52}$$

with the upper boundary $r_{\text{ref}}(0)$ being the stretched out configuration of the robot. This defines

$$\mathcal{A}_4 := \left\{ (\boldsymbol{p}, \lambda) \in \mathcal{P} \ \middle| \ \sqrt{r_{SE}^2 + r_{EW}^2 - 2r_{SE}r_{EW}\cos\left(\pi - q_4^{\max}\right)} \leq r_{\text{ref}} \leq r_{SE} + r_{EW} \right\} \tag{53}$$

as the admissible parameter set w.r.t. joint 4.

3.5.2. Null Space Parameter $\lambda$

All remaining joints, i.e., shoulder joints $\{q_1, q_2, q_3\}$ and wrist joints $\{q_5, q_6, q_7\}$, limit parts of the null space parameter $\lambda$. The 4-quadrant $\text{atan2}(\cdot)$ functions from (43), however, are difficult to

symbolically rewrite in terms of $\lambda$ due to there piecewise definition. To circumvent this, we further introduce IK mappings that calculate the absolute joint angles. We define the extraction map of absolute values of the Euler sequence $|\operatorname{eul}_{ZYZ}| : \mathrm{SE}(3) \to \mathbb{R}^3_+$ as

$$
|\operatorname{eul}_{ZYZ}| : \quad \mathrm{SE}(3) \to \mathbb{R}^3_+, \quad A_{zyz} \mapsto
\begin{bmatrix}
\arccos\left( \dfrac{\left[ A_{zyz}(z) \right]_{(3,1)}}{\sin\left( \arccos\left( \left[ A_{zyz}(z) \right]_{(3,3)} \right) \right)} \right) \\[12pt]
\arccos\left( \left[ A_{zyz}(z) \right]_{(3,3)} \right) \\[12pt]
\arccos\left( \dfrac{-\left[ A_{zyz}(z) \right]_{(1,3)}}{\sin\left( \arccos\left( \left[ A_{zyz}(z) \right]_{(3,3)} \right) \right)} \right)
\end{bmatrix}
=:
\begin{bmatrix}
|\gamma| \\[12pt]
|\beta| \\[12pt]
|\psi|
\end{bmatrix}
\tag{54}
$$

which is used to find the absolute angles of the shoulder and wrist joints

$$
\begin{bmatrix}
|q_1 + \gamma_{\mathrm{ref}}| \\
|q_2| \\
|q_3|
\end{bmatrix}
(p, \lambda) := |\operatorname{eul}_{ZYZ}|(A_{3B}(p, \lambda))
\tag{55a}
$$

$$
\begin{bmatrix}
|q_5| \\
|q_6| \\
|q_7 + \psi_{\mathrm{EE}}|
\end{bmatrix}
(p, \lambda) := |\operatorname{eul}_{ZYZ}|(A_{74}(p, \lambda))
\tag{55b}
$$

analogously to the mapping $\operatorname{eul}_{ZYZ}$ from the previous Section 3.3. See Appendix C for the full definition of the absolute valued IK functions. Note that the mappings admit the same symmetrical assignments between the shoulder and wrist portion as the actual IK mapping discussed in Section 3.3.5.

Due to the concise formulations of the IK (55a), all functions can be solved for the null space parameter $\lambda$. By substituting the joint parameters with their respective limit, closed-form expressions are formed that deliver $s_i$ candidates for lambda ranges

$$
\lambda_i^{\lim} : \quad \mathbb{R}^7 \times \mathbb{R} \to \mathbb{C}^{s_i}, \quad (p, q_i^{\max}) \mapsto \lambda_i^{\lim}(p, q_i^{\max}) \quad \forall i \in [1,7] \setminus 4
\tag{56}
$$

according to the $i = [1,7]$ joints. For $q_2$ and $q_6$, the respective middle joints of the shoulder and wrist angle tuples $(q_1, q_2, q_3)$ and $(q_5, q_6, q_7)$, we directly find $s_2 = s_6 = 2$ symmetric solutions for a positive and negative null space limit. However, solving the remaining mappings from IK (55a) for $\lambda$, results in more solution candidates. This results from the fact that, depending on the parameter configuration, these joints have the potential for cyclic behaviour for a linear increase in $\lambda$ at a fixed pose (discussed in [15]). Joints $q_3$ and $q_5$ can thus reach up to $s_3 = s_5 = 4$ null space angles marking a joint limit. The first joint $q_1$ and last joint $q_7$ do also offer up to 4 critical values for $\lambda$, however, due to additional additive parameters $\gamma_{\mathrm{ref}}$ and $\psi_{\mathrm{EE}}$ resp., it is necessary to additionally consider solutions for $|-q_1 + \gamma_{\mathrm{ref}}|$ and $|-q_7 + \gamma_{\mathrm{ref}}|$. These solutions are evaluated with $\lambda_1^{\lim}(p|_{-\gamma_{\mathrm{ref}}}, q_1^{\max})$ and $\lambda_7^{\lim}(p|_{-\psi_{\mathrm{EE}}}, q_7^{\max})$. Consequently, $s_1 = s_7 = 8$ solution candidates for the first and last joint of the kinematic have to be considered.

Besides knowing the value of a critical limit, it is further essential for many applications to know if it expresses an upper or a lower limit. Similar to the approach in [16], the partial derivatives of the null space range mappings $\lambda_i^{\lim}$ w.r.t. the corresponding joint angle limit are used to characterize each limit candidate. For every $\ell \in \lambda_i^{\lim}$, the corresponding partial derivative is evaluated to decide

$$
\ell \in
\begin{cases}
\text{is upper limit} & \text{if } \operatorname{sign}(\ell)\, \dfrac{\partial \lambda_i^{\lim}}{\partial q_i^{\max}} > 0 \\[10pt]
\text{is lower limit} & \text{if } \operatorname{sign}(\ell)\, \dfrac{\partial \lambda_i^{\lim}}{\partial q_i^{\max}} < 0 \\[10pt]
\text{is no limit} & \text{otherwise.}
\end{cases}
\tag{57}
$$

In a second step, all solution candidates in $\lambda_i^{\lim}$ are tested for validity, to define the sets of actual upper and lower null space limit angles

$$\mathcal{L}_i^{\mathrm{up}}(\boldsymbol{p}) := \left\{ \lambda \in \boldsymbol{\lambda}_i^{\lim} \;\middle|\; \lambda \in \mathbb{R} \;\wedge\; |\mathrm{IK}_i(\boldsymbol{p},\lambda)| = q_i^{\max} \;\wedge\; \mathrm{sign}\,(\lambda)\,\frac{\partial \lambda}{\partial q_i^{\max}} > 0 \right\} \quad \forall i \in [1,7] \setminus 4 \quad \text{(58a)}$$

$$\mathcal{L}_i^{\mathrm{low}}(\boldsymbol{p}) := \left\{ \lambda \in \boldsymbol{\lambda}_i^{\lim} \;\middle|\; \lambda \in \mathbb{R} \;\wedge\; |\mathrm{IK}_i(\boldsymbol{p},\lambda)| = q_i^{\max} \;\wedge\; \mathrm{sign}\,(\lambda)\,\frac{\partial \lambda}{\partial q_i^{\max}} < 0 \right\} \quad \forall i \in [1,7] \setminus 4. \quad \text{(58b)}$$

These upper and lower limits form $j$ pairwise ranges $\Lambda_{i,j}$ and define the remaining admissible parameter sets

$$\mathcal{A}_i := \left\{ (\boldsymbol{p},\lambda) \in \mathcal{P} \;\middle|\; \lambda \in \bigcup_j \Lambda_{i,j} \right\} \quad \forall i \in [1,7] \setminus 4 \,, \tag{59}$$

related to shoulder and wrist joints.

The full intersection set $\mathcal{A}$, as defined in (50), may consist of several separate regions. Directly evaluating all critical values of $\lambda$ is especially interesting whenever planning a continuous path in task space. We apply the admissible parameter space in application Sections 5.1.3 and 5.2.2. All full function definitions of the limit candidates $\lambda_i^{\lim}$ are summarized in Appendix D.

## 4. Results

This section contains an evaluation of the task space manipulability framework developed in this work. We first give a run-time comparison to show the computational advantage of our closed-form expression in comparison to general numerical solutions. We show that uniform sampling in the new parameter space results in a superior probability distribution of the manipulability in comparison with direct sampling in joint space. Further, the sensitivity of the manipulability measure w.r.t. the parameters is analyzed.

### 4.1. Accuracy

Unlike numerical IK solvers that approximate the inverse mapping iteratively [42], or CLIK solvers [10–12] that converge to the solution from a control point of view, the analytical nature of our closed-form task space manipulability expression delivers exact results in a single iteration.

### 4.2. Run-Time Comparison

Complete evaluation of the closed-form IK and M mapping as single expressions allows automatic code generation of the symbolic expressions with e.g., the *MATLAB Coder™* toolbox. These expressions allow array operations, or *vectorization* in *MATLAB*, such that a large number of solutions can be evaluated simultaneously. This leads to a significant computational boost, compared to algorithms that rely on matrix arithmetic and consequently have to sequentially evaluate multiple evaluations in programmatic loops. This property makes it further straightforward to calculate the task space manipulability of multiple samples on a powerful Graphics Processing Unit (GPU). The following run-time comparison was conducted in *MATLAB 2019a*, on a computer with Intel(R) Core(TM) i9-9900X CPU @ 3.50 GHz, 128 GB memory, and a *NVIDIA TITAN V* graphics card.

Besides different versions of our presented algorithm, we also tested the run-time of [15], representing typical analytical IK approaches in the literature, and the nonlinear optimization-based IK algorithm from the *Robotics System Toolbox™* for *MATLAB*, representing iterative solver approaches. Figure 5 shows a run-time comparison of calculating the manipulability measures

$$\mu_n := (\mathrm{M} \circ \mathrm{IK})(\boldsymbol{p}_n), \quad \text{for } n = [1,N] \tag{60}$$

of $N$ random samples $\boldsymbol{p}_n$.

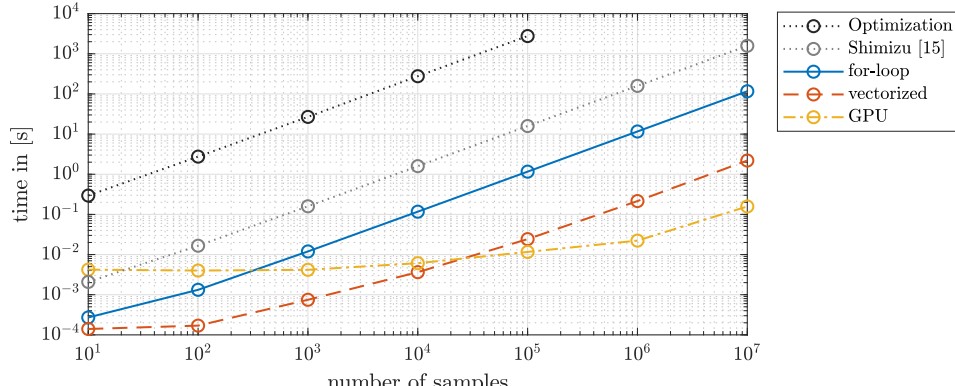

**Figure 5.** Run-time comparison of processing *N* poses w.r.t. their task space manipulability. Considered are the *MATLAB* robotics IK solver based on nonlinear optimization, the analytical IK solver by Shimizu et al. [15], and the presented approach in three versions: a conventional sequential loop structure, as well as vectorized evaluation on the central processing unit (CPU) and graphics processing unit (GPU).

As expected, the iterative optimization algorithm (applying BFGS Gradient Projection with solution tolerance 0.01) is the computationally most expensive solution method. It required an average of 37 iterations per pose and did not allow for direct selection of the arm angle. Two orders of magnitude faster and in addition producing exact inverse solutions are the analytical IK solvers found in the literature. They rely on matrix calculus and thus a for-loop structure for evaluation of multiple poses.

Our approach, which is entirely reduced to direct individual expressions, is over 10 times faster when implemented with the same conventional for-loop structure. Already for 200 evaluated samples, a simultaneous vectorized evaluation achieves another performance increase of factor 10. At the maximal evaluated amount of $10^7$ samples, vectorization enables an even 50 times faster computation, compared to the implementation using for-loops. The advantage of calculating the task space manipulability on a GPU starts at an amount of $10^5$ sample points. For a smaller number of samples, the overhead of initializing the data on the GPU does not pay off. Processing $10^7$ samples, calculations on the GPU are 10 times faster then vectorized treatment on the CPU, and even 700 times faster than for conventional loop structures. Note that all time measurements include the generation of random samples on the CPU and GPU respectively.

Considering real-time application for a robot with a typical 1 kHz sampling rate, our approach allows evaluation of 1000 end-effector poses for their task space manipulability.

### 4.3. Sampling in Task Space

Not having to calculate the IK in an iterative fashion as done by CLIK solvers, evaluating manipulability directly in task space is computationally not much more expensive than directly calculating manipulability in joint space. However, choosing a different space for sampling random poses do have an influence on the probability distribution of resulting manipulability measures.

Before analyzing this difference, we first discuss the used sampling strategies. For a fair comparison, we cover the entire space without consideration of possible limits on the individual joints or parameters.

Let $u \in \mathbb{R}$ be a random number drawn from a uniform distribution in the range of $[0, 1]$. Uniform sampling in joint space is straightforward with

$$q_i^{\text{uniform}} : \quad \mathbb{R} \to \mathbb{R}, \quad u \mapsto -\pi + 2\pi u \qquad \forall i \in [1, 7] \tag{61}$$

due to the independence of its joints $q \in \mathbb{R}^7$.

For a random end-effector pose sample $(\boldsymbol{p}^{\text{red}}, \lambda) = [r_{\text{ref}}, \beta_{\text{ref}}, \gamma_{\text{EE}}, \beta_{\text{EE}}, \lambda]^{\top}$ from the parameter space, one can choose the same strategy

$$p_1^{\text{naive}}: \quad \mathbb{R} \to \mathbb{R}, \quad u \mapsto r_{\text{ref}}^{\min} + (r_{\text{ref}}^{\max} - r_{\text{ref}}^{\min})u \tag{62a}$$

$$p_i^{\text{naive}}: \quad \mathbb{R} \to \mathbb{R}, \quad u \mapsto -\pi + 2\pi u \quad \forall i \in [2, 5] \tag{62b}$$

with respective scaling for the linear parameter $r_{\text{ref}}$. However, this *naive* form of sampling does not lead to a uniform distribution of samples in the task space SE(3), due to the interdependence of the coordinate components.

Recall that the first two parameters $r_{\text{ref}}$ and $\beta_{\text{ref}}$ describe translation in polar coordinates. Unlike in Cartesian coordinates, the base vectors are not constant. Consequently, direct uniform sampling of the radial coordinate $r_{\text{ref}}$, leads to sparser sampling further from the origin, due to the increasing circumference proportionally to $r_{\text{ref}}$. Proper uniform sampling of the translational part can be achieved by

$$p_1^{\text{uniform}}: \quad \mathbb{R} \to \mathbb{R}, \quad u \mapsto \sqrt{\left(r_{\text{ref}}^{\min}\right)^2 + \left(\left(r_{\text{ref}}^{\max}\right)^2 - \left(r_{\text{ref}}^{\min}\right)^2\right) u} \tag{63a}$$

$$p_2^{\text{uniform}}: \quad \mathbb{R} \to \mathbb{R}, \quad u \mapsto -\pi + 2\pi u. \tag{63b}$$

An efficient method of uniform sampling on SO(3), i.e., 3D orientations, is proposed by Kuffner [43]. Uniform sampling of the individual angles of the Euler sequence results in a bias towards the polar regions of the unit sphere. He proposes to use an arctan function on the second angle to compensate for this bias. Uniform sampling of the end-effector orientation, parametrized by $\gamma_{\text{EE}}$ and $\beta_{\text{EE}}$, is thus achieved with

$$p_3^{\text{uniform}}: \quad \mathbb{R} \to \mathbb{R}, \quad u \mapsto -\pi + 2\pi u \tag{63c}$$

$$p_4^{\text{uniform}}: \quad \mathbb{R} \to \mathbb{R}, \quad u \mapsto \arccos\left(1 - 2u\right). \tag{63d}$$

The last portion in our parameter tuple $(\boldsymbol{p}, \lambda)$ is the null space parameter $\lambda$ that is independent and thus remains

$$p_5^{\text{uniform}}: \quad \mathbb{R} \to \mathbb{R}, \quad u \mapsto -\pi + 2\pi u. \tag{63e}$$

Figure 6 illustrates the uniform sampling of the task space applying the uniform sampling strategy (63).

The above-discussed sampling strategies are now analyzed in conjunction with their respective mapping to the manipulability measure. Figure 7 shows the approximated cumulative distribution function (CDF) of manipulability resulting from $10^7$ random samples. It shows that random sampling in joint space according to (61) is more likely to result in a joint configuration with poor manipulability of the robot. Uniform sampling in parameter space (63) produces much fewer joint configurations with poor manipulability, while at the same time more configurations with high manipulability. Naive sampling in parameter space (61) performs similarly good in the low manipulability section. However, it produces also fewer configurations with high manipulability. Considering a conventional 6-DOF robot, i.e., fixing the null space parameter $\lambda$ to 0 or $\pi$, results in a slightly better probability density function (PDF) than for the discussed 7-DOF mechanism. This is a surprising result, as it is always argued that the redundancy improves manipulability. While it is true that the additional DOF has the potential to improve performance measures, poor exploitation might achieve the opposite.

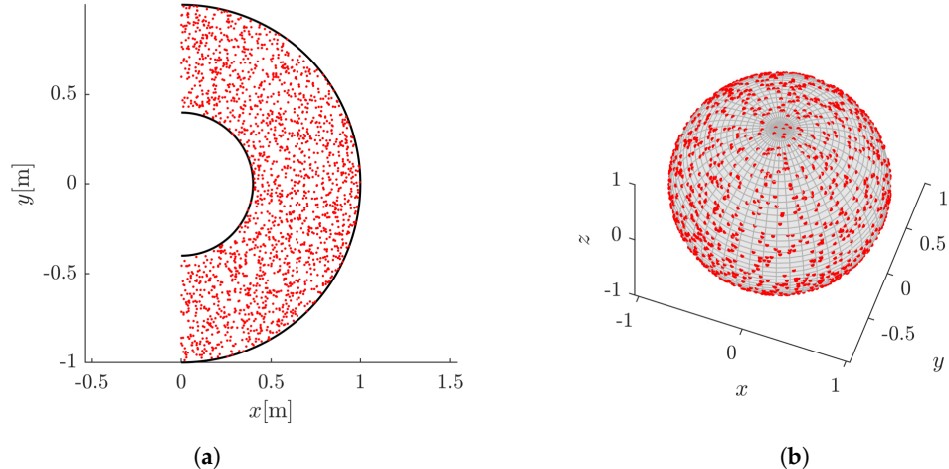

**Figure 6.** Uniform distributed sampling of the task space (2000 samples). (**a**) End-effector translation; (**b**) End-effector orientation.

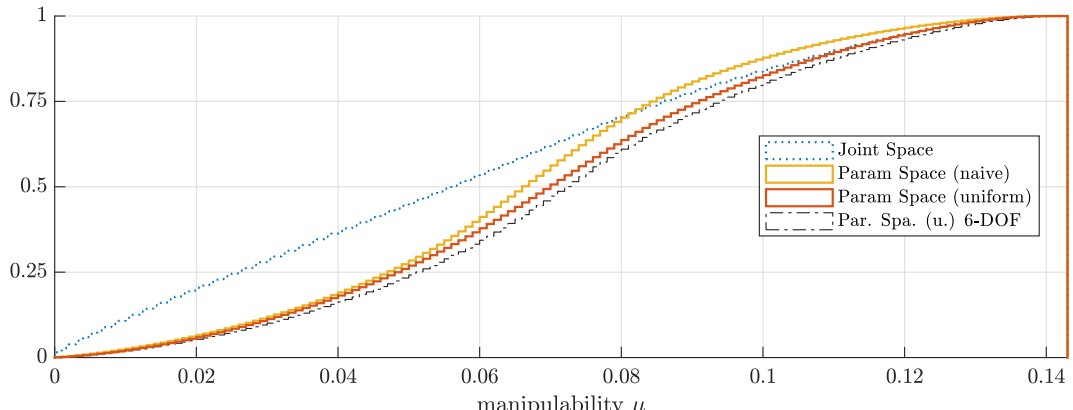

**Figure 7.** Approximated cumulative distribution function (CDF) from a histogram of manipulability w.r.t. different sampling strategies ($10^7$ samples).

Kuhlemann et al. [44] showed in different use-cases that the seventh DOF of the *KUKA LBR iiwa* increased the average dexterity by 16% in comparison to a conventional 6 DOF *KUKA KR 10*. Both the shortcomings of the naive parameter sampling strategy and the apparent advantage of the 6-DOF mechanism are discussed in Section 4.4.4.

The average normalized manipulabilities achieved are 37% for uniform joint space sampling, 43% for naive parameter space sampling, and 50% for uniform sampling in parameter space. All numbers are w.r.t. the maximal encountered manipulability.

### 4.4. Parameter Sensitivity Analysis of Manipulability in Parameter Space

The sensitivity of the task space manipulability w.r.t. its parameters are analyzed by generating $10^7$ random samples according to (63). These samples represent a uniform distribution of task space configurations. Figure 8 shows the bi-variate histograms of manipulability $\mu(\boldsymbol{p}^{\mathrm{red}}, \lambda)$ w.r.t. to the individual parameters.

Colors approximate the PDF of $\mu(r_{\mathrm{ref}}, \beta_{\mathrm{ref}}, \gamma_{\mathrm{EE}}, \beta_{\mathrm{EE}}, \lambda)$ at fixed values of the respective parameter. For all parameter values we find unimodal distributions, i.e., distributions with a single maxima.

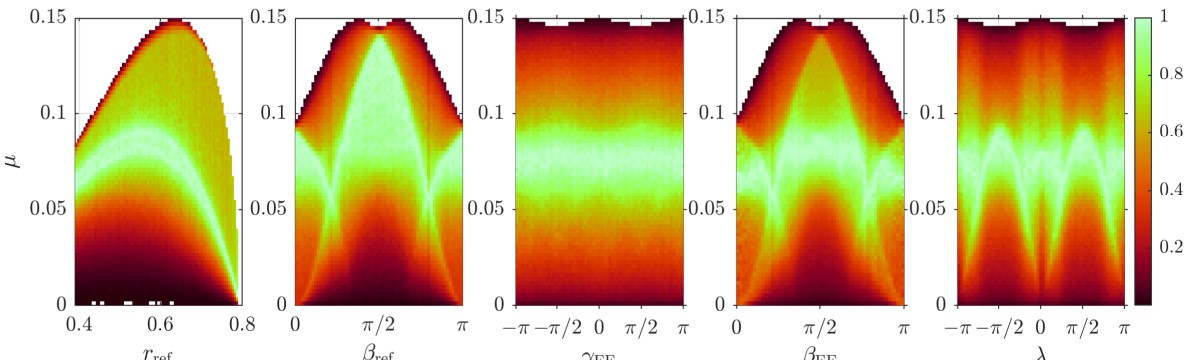

**Figure 8.** Bi-variate histograms of $\mu(r_{\text{ref}}, \beta_{\text{ref}}, \gamma_{\text{EE}}, \beta_{\text{EE}}, \lambda)$ w.r.t. to the individual parameters, based on $10^7$ uniformly distributed parameter space samples. Colors are normalized along with the particular value of the parameter on the x-axis.

### 4.4.1. Translation Parameters $r_{\text{ref}}$ and $\beta_{\text{ref}}$

The PDF of $\mu$ along the shoulder-wrist distance $r_{\text{ref}}$ shows a preferred value of 0.57 m. Although a manipulability optimizing configuration cannot be found at this given value, the mode of the corresponding PDF, i.e., its local maxima, has the highest value of manipulability. Further, the probability of good manipulation is decreasing with $r_{\text{ref}}$ towards the workspace singularity, i.e., a fully stretched arm of robot configuration.

The polar angle $\beta_{\text{ref}}$ between the vertical and the shoulder-wrist reference vector has the highest manipulability mode at $\frac{\pi}{2}$ rad, although manipulability maximizing configurations are not found. For values approaching 0 and $\pi$ rad, i.e., placing the wrist in line with the axis of base joint $q_1$, typically cause so-called *shoulder singularities* on conventional 6-DOF robots. While the 7-DOF kinematics do not necessarily result in a kinematic singularity, high manipulability is not possible either.

### 4.4.2. Orientation Parameters $\gamma_{\text{EE}}$ and $\beta_{\text{EE}}$

The third parameter $\gamma_{\text{EE}}$, which describes a rotation around the shoulder-wrist vector, is the only one that seems to cause little variation in the manipulability PDF and does not allow a conclusion over a preferred configuration.

The consecutive rotation angle $\beta_{\text{EE}}$ shows a similar influence as the reference angle $\beta_{\text{ref}}$. However, the mode of these PDFs is less prominent and tendentiously marks a lower manipulability.

### 4.4.3. Null Space Parameter $\lambda$

The null space parameter $\lambda$ reveals that the highest manipulabilities can be found at $\lambda = \{0, \pm\pi\}$rad, i.e., the conventional upper and lower elbow configuration of 6-DOF kinematics. Although missing the absolute top manipulability poses, only small deviations of about $\pm 0.1$ rad from these configurations result in a decrease of the manipulability mode of 25%, i.e., from 0.8 to 0.6. Better modes are found at $\lambda = \{\pm\frac{\pi}{2}\}$rad. Not only is their peak at a slightly higher manipulability of 0.85, but they are also less sensitive to a parameter change in $\lambda$. The latter is especially valuable for staying agile during unforeseen events.

### 4.4.4. Discussion of Manipulability in Different Sampling Strategies

The different sampling strategies discussed in Section 4.3 result in differences in the approximated CDFs, cf. Figure 7.

Naive vs. Uniformly Distributed Sampling

The difference between naive and uniform sampling solely affects parameters $r_{\text{ref}}$ and $\beta_{\text{EE}}$. That is, the corresponding uniform sampling functions (63a) and (63d) correct the biases of the radial

coordinate $r_{\text{ref}}$ towards the origin, and the orientation towards the pole regions with azimuthal angle $\beta_{\text{EE}} = \{0, \pi\}$rad, respectively. Consequently, these regions are sparser sampled in the uniformly distributed strategy. While this correction is negligible for the range of $r_{\text{ref}}$ in this particular robot example, the improvement of the CDF towards better manipulability stems from a sparser sampling of the boundary regions of $\beta_{\text{EE}}$. Because exactly these boundaries lack high manipulability poses, as visible in the according bi-variate histogram in Figure 8.

6-DOF vs. 7-DOF Kinematics

According to Section 4.4.3, the apparent slight advantage of uniform distributed sampling of a conventional 6-DOF robot only holds for the over-all manipulability distribution illustrated in Figure 7. The parameter-specific histogram w.r.t. to the arm angle $\lambda$ in Figure 8, on the other hand, reveals that the conventional 6-DOF configurations $\lambda = \{0, \pm\pi\}$rad do have a good manipulability distribution, but $\lambda = \{\pm\frac{\pi}{2}\}$rad configurations are preferable. A 7-DOF kinematics hence not only enables agile adaptation of the kinematic structure, but also contains arm angles that have a better PDFs of manipulability than its 6-DOF counterpart. At the same time, other arm angles show higher variability in the histogram and are more prone to decrease performance. An increase in manipulability by the additional DOF thus relies on a well-conceived utilization of such.

*4.5. Number of Local Optima*

While the analysis shown in the previous section gives insight in the probability distribution of the manipulability measure, it does not allow conclusions on how manipulability changes along the null space. Table 1 lists the number of local optima for a given end-effector pose. It shows that 80% of the robot poses do not have a unique manipulability maximizing null space solution, but up to 4 distinct optima.

**Table 1.** Distribution of local optima among $10^7$ samples.

| # optima | 1 | 2 | 3 | 4 |
|---|---|---|---|---|
| percentage | 20% | 41% | 27% | 12% |

## 5. Applications

Two application directions that benefit from the closed-form expressions of the task space manipulability are outlined in this section. First, we demonstrate how global optimization problems can be formulated that profit from massive multi-start point pre-evaluation. Second, we propose a novel way of real-time redundancy resolution on the position level, which enables global manipulability optimization of single poses as well as for provided end-effector trajectories in SE(3).

*5.1. Optimal Robot Placement*

The analytic results from the previous Section 3 allow formulating interesting questions in terms of optimization problems. We consider the problem of optimal placement of the robot.

5.1.1. Best Overall Robot Configuration

The most basic optimization problem we considere is the question of finding the best overall robot configuration w.r.t. to manipulability. Mathematically, this problem can be stated as an unconstrained optimization problem

$$\underset{q}{\text{maximize}} \quad \mu(\boldsymbol{q}) \tag{64}$$

directly finding the optimal joint configuration w.r.t. the manipulability measure. The global optimum is found with a multi-start strategy [45], where random samples are drawn from the admissible

parameter space $\mathcal{P}$ and used as starting points for local optimizations. Figure 9, left side, shows the results of such a global optimization process with 1000 starting points. Note that the same problem can be formulated in *parameter space* and does yield the same result. All optimization iterations result in one of 8 equally good global optima, which can be reduced to 4 solutions due to symmetry of the shoulder joint. They further describe configurations in the pure $xz$-plane with $\lambda \in \{0, \pm 180\}°$. This is equivalent to the configurations achievable by a conventional 6-DOF robot.

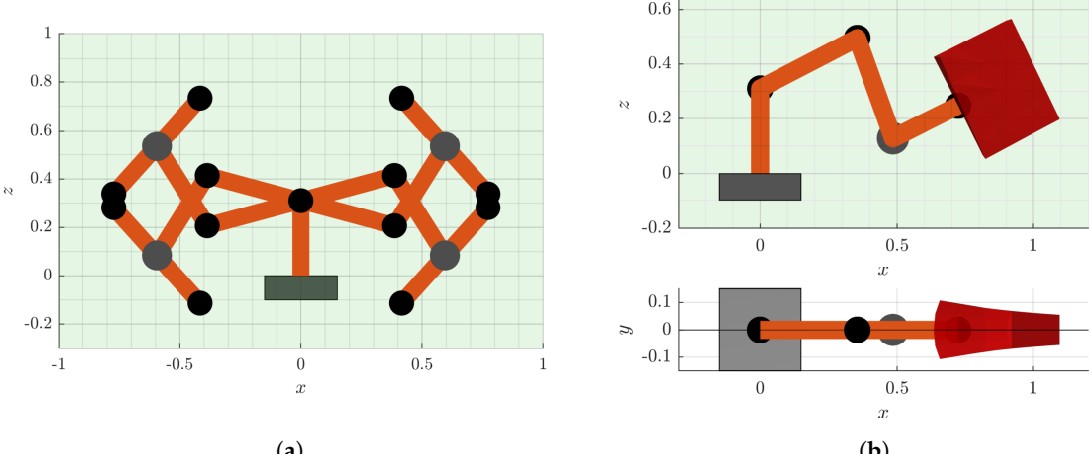

(**a**)                    (**b**)

**Figure 9.** Results of the task space manipulability optimization of a robot mounting pose. (**a**): Overall best robot configuration. There are a total of 8 global optima with equal manipulability $\mu_{\max} = 0.143$. From 1000 random initial starting points, 83% of the optimization runs converged to one of the global optima. (**b**): Optimizing relative pose w.r.t. a workspace envelope of size $(\Delta x, \Delta y, \Delta z) = (0.4, 0.4, 0.3)$m. Note that the cubic volume is projected onto the parameter space, hence the distortion in the illustration. The resulting configuration for pose $z_0$, again lies fully on the $xz$-plane. However, unlike the single best pose, only one single optimum is found.

### 5.1.2. Best Robot Configuration for Multiple Task Poses

In industrial settings, robots are often required to work at a certain number $i \in \mathbb{Z}^+$ of different task poses $z_i$. While the relative distances $\Delta z_i = z_i - z_1$ between this poses is defined, the optimal placement of the robot can be found by solving the optimization problem

$$\underset{z, \lambda}{\text{maximize}} \quad \sum_i (\text{M} \circ \text{IK} \circ \text{TSP})(z + \Delta z_i, \lambda) \tag{65}$$

to find the relative pose $z$ that maximizes the average manipulability of all $i$ poses. Solving this problem directly, results in an infinite number of global poses. These solutions are rotationally symmetric around the base joint $q_1$ as well as the last joint $q_7$, as both these joints do not have an influence on the manipulability of the 7-DOF robot structure at consideration (discussed in Section 3.1).

The complexity of the optimization problem, as well as the number of global optima, can be drastically reduced by formulating the same problem in the lower dimensional parameter space

$$\underset{p, \lambda}{\text{maximize}} \quad \sum_i (\text{M} \circ \text{IK})(p + \Delta p_i, \lambda) \tag{66}$$

where $p_i = \text{TSP}(z_i)$. The resulting optimal $p$ can eventually be mapped to the corresponding task space parameter $z = \text{TSS}(p)$. This result is useful for deciding on how to mount a robot relative to a given set of task poses $z_i$, or recalculating it online if task poses are time-variant and the robot structure is e.g., mounted on a mobile platform.

### 5.1.3. Optimizing Robot Mounting Positions Regarding a Workspace Envelope

In a modern scenario where robots are not only expected to repetitively execute the same tasks, a set of pre-defined task poses cannot always be formulated. But it is rather necessary for the robot to perform well in a defined workspace volume, e.g., given as a cubical volume $V = [-\Delta\frac{x}{2}, +\Delta\frac{x}{2}] \times [-\Delta\frac{y}{2}, +\Delta\frac{y}{2}] \times [0, +\Delta z]$. Due to all mappings involved in the task space manipulability being continuous, formulating a cost function for such a volume can be done using Fubini's theorem [46]. It allows calculation of the volume integral as triple integral. The objective for this optimization problem in task space reads

$$\underset{z_0, \lambda}{\text{maximize}} \iiint\limits_V (\text{M} \circ \text{IK} \circ \text{TSP})(z_0 + z(x,y,z), \lambda) \, dx \, dy \, dz \tag{67}$$

$$\text{subject to} \quad \text{TSP}\,(z_0, +z(x,y,z), \lambda) \in \mathcal{A},$$

where the optimal task space volume origin $z_0$ needs to be found. This optimization can again be transformed to the lower dimensional parameter space

$$\underset{p_0, \lambda}{\text{maximize}} \iiint\limits_V (\text{M} \circ \text{IK} \circ \text{TSP})(\text{TSS}(p_0) + z(x,y,z), \lambda) \, dx \, dy \, dz \tag{68}$$

$$\text{subject to} \quad \text{TSP}\,(\text{TSS}(p_0, \lambda) + z(x,y,z), \lambda) \in \mathcal{A}$$

with the condition that the whole Volume projected to parameter space must be within the set of admissible parameters. Figure 9, right side, shows the result of such a global optimization.

### 5.2. Redundancy Resolution

Solving for optimal robot poses online is essential for a robot to stay agile at all times. We demonstrate how the task space manipulability expressions developed in this work can be applied for real-time global manipulability optimization of single poses as well as full trajectories. The following run-time evaluations were conducted in *MATLAB 2019a*, on a computer with Intel(R) Core(TM) i3-7100 CPU @ 3.9 GHz and 32 GB memory.

### 5.2.1. Redundancy Resolution for Global Manipulability Optima

Approaches typically found in the literature focus on local optimization of manipulability based on local gradient information. Analysis of the number of existing local optima from Section 4.5, however, revealed that only 20% of end-effector poses have a unique global optimum. The computational advantage of our approach permits evaluating the manipulability of many poses simultaneously. Given a current robot pose $z$, our framework makes it possible to not only locally improve manipulability, but solve

$$\underset{\lambda}{\arg\max} \quad (\text{M} \circ \text{IK} \circ \text{TSP})(z, \lambda) \tag{69}$$

with a representative number of null space solution at a high resolution in real-time. Given the information of this *greedy optimization* strategy, the close-to-global optimum configuration can simply be picked. Solving for global optima in 0.25 ms at a resolution of 1° for $\lambda$ enables application at typical robot sampling rates of 1 kHz.

Figure 10 shows manipulability of the full null space at a particular configuration.

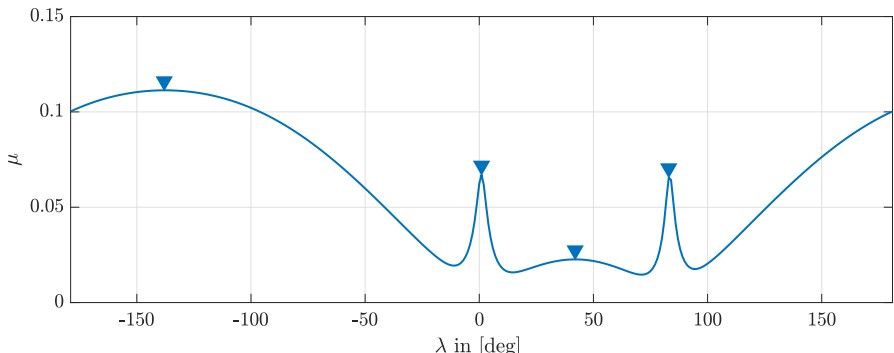

**Figure 10.** Multiple local optima of manipulability $\mu$ in the null space of $\boldsymbol{p}^{\text{red}} = [0.6, 0.7, 1.4, 0.7]^{\top}$.

This is an example of a pose with 4 local optima. If the current configuration of the robot is the solution for the given pose with the null space parameter $\lambda \in [0, 85]°$, a local optimization will only drive the redundancy resolution into a sub-optimal minima. In contrast, our approach allows finding the globally best configuration w.r.t. the admissible parameter space.

### 5.2.2. Optimizing Null Space Solution of Given End-Effector Trajectory

Several approaches can be found in the literature that maximize either the volume of a manipulability ellipsoid [31,47–49] or a predefined shape of the ellipsoid [33]. Yet all these approaches consider only local optimization.

Finding the best joint configuration for a given pose in task space simplifies to a 1D line search. However, given a full path in SE(3) it is also possible to find an optimal elbow trajectory that maximizes, e.g., the average manipulability while avoiding getting trapped in regions of poor manipulability. Note that a real manipulation task relies on a sophisticated path planner, capable of generating task-related paths that avoid obstacles while potentially fulfilling additional criteria. Knowledge about the task space manipulability, e.g., provided by our approach, may even be exploited by such a planner. This is, however, not the direct scope of this work. Instead, for a minimal working example, we use direct interpolation

$$\boldsymbol{p}(s) = s\boldsymbol{p}^{\text{start}} + (s-1)\boldsymbol{p}^{\text{end}} \quad \text{with} \quad s = [0,1] \tag{70}$$

between two poses as a simple path planner. Given are two random poses as depicted in Figure 11 to the left. On the right side of Figure 11, a contour plot of the manipulability of the full null space along the trajectory is shown. Red lines indicate not passable values in the null space due to joint limits, cf. Section 3.5. The blue line marks the trajectory that results from local optimization of manipulability. Note that at $s = 0.4$, the local optimization hits a joint limit of $q_2$. We stopped the line here, because it depends on a potential strategy for joint limit avoidance, which is not the scope of this work. A global optimization strategy that has predictive knowledge of the full null space development can exploit an initially sub-optimal path toward negative values of $\lambda$ to circumvent the region of poor manipulability between $s = [0.6, 1]$. But this is usually not feasible in an online scenario with conventional global optimization strategies.

The computational advantage of our strategy, as seen in Figure 5, allows the computation of such a map with, e.g., a resolution of 100 steps in both parameters, $s$ and $\lambda$, in under 5 ms. In combination with an online trajectory generator directly on SE(3), e.g., [50], this qualifies our task space manipulability approach to be used for predictive online manipulability optimization, e.g., with a receding horizon.

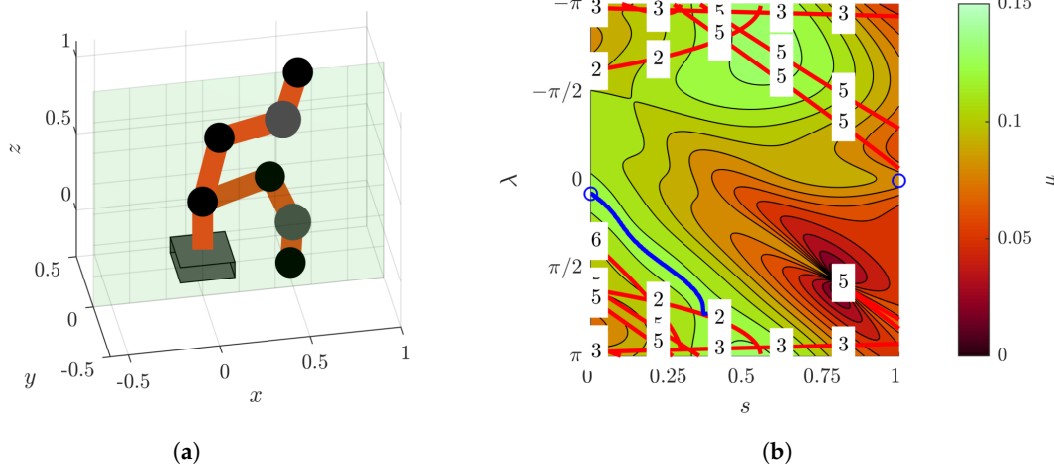

(**a**)          (**b**)

**Figure 11.** Null space manipulability over a parameter trajectory. The 3D plot (**a**) shows an exemplary start $p(s = 0)$ and end configuration $p(s = 1)$. The contour plot (**b**) shows the manipulability $\mu(p, \lambda)$ of the full null space. Red lines mark the limits $\lambda(p, q_{max})$ of the admissible null space region. The numbers refer to the invoking joint. Blue circles mark desired $\lambda^d(s = 0)$ and $\lambda^d(s = 1)$, and the blue line marks a trajectory as it would be chosen by local optimization of $\lambda_{max}(p)$.

## 6. Conclusions

Today's demand for adaptive and reactive robot behaviour requires sustaining the agility of a kinematic structure at all times. While manipulability is a common metric in robot research to quantify the capabilities of a robot at a given joint configuration, the robot task is directly defined in end-effector poses, which allows for multiple possible solutions. Unlike common metrics, which do not include the robot IK, a task space manipulability formulation is required to directly map an end-effector pose together with its null space solution onto the manipulability metric.

To achieve reactive robot behaviour, optimization of the null space at given poses must be performed online. In general, this requires efficient evaluation of a large number of configurations, especially in the case of redundant robots. In this work we developed a new closed-form approach for calculating manipulability directly from task space poses, for a redundant 7-DOF S-R-S serial robot kinematics. A novel parametrization of the task- and null space leads to concise IK, as well as admissible parameter mappings, which show symmetry in the structures of their individual expressions. Analysis of the resulting task space manipulability further revealed that the majority of end-effector poses do not have a unique, manipulability-maximizing null space solution. We thus argue that local optimization of the manipulability measure is not sufficient. A global optimization at high sampling frequencies, however, is not feasible with current approaches in the literature. The entire composition of the task space manipulability map proposed in this work allows for efficient array operations that can be exploited in vector-optimized programming languages, as well as GPU computing. Consequently, the simultaneous computation of a large number of poses in real-time is made possible. Our method, therefore, enables global online optimization of manipulability for single poses and even full SE(3) trajectories.

Future work will focus on further application development of our framework. Combining our task space manipulability approach with online planners opens an interesting field of predictive redundancy resolution for global manipulability optimization.

**Author Contributions:** conceptualization, G.H.; methodology, G.H.; software, G.H.; validation, G.H.; formal analysis, G.H.; investigation, G.H.; resources, D.W.; data curation, G.H.; writing–original draft preparation, G.H. and D.W.; writing–review and editing, G.H. and D.W.; visualization, G.H.; supervision, D.W.; project administration, D.W.; funding acquisition, D.W.,

**Funding:** The research leading to these results has received funding from the Horizon 2020 research and innovation programme under grant agreement №820742 of the project "HR-Recycler - Hybrid Human-Robot RECYcling plant for electriCal and eLEctRonic equipment".

**Acknowledgments:** The authors would like to thank Adam Horvath, Thomas Hartmann and Christian Ritter for discussions and initial implementations of the results.

**Conflicts of Interest:** The authors declare no conflict of interest. The funders had no role in the design of the study; in the collection, analyses, or interpretation of data; in the writing of the manuscript, or in the decision to publish the results.

## Abbreviations

The following abbreviations are used in this manuscript:

| | |
|---|---|
| CDF | cumulative distribution function |
| CLIK | Closed-Loop Inverse Kinematic |
| CPU | Central Processing Unit |
| DOF | degree of freedom |
| FK | Forward Kinematic |
| GPU | Graphics Processing Unit |
| IK | Inverse Kinematic |
| M | Manipulability |
| PDF | probability density function |
| PR | parameter requirements |
| SIMD | Single Instruction Multiple Data |
| S-R-S | Spherical-Revolute-Spherical |
| TCP | tool center point |
| TSP | Task Space Projection |
| TSS | Task Space Surjection |

## Appendix A. Manipulability

The full manipulability map M, discussed in Section 3.1, is given with $\mu(\boldsymbol{q}) = \sqrt{\det\left({}_T\boldsymbol{J}\,{}_T\boldsymbol{J}^\top\right)}$. Note that the Jacobian matrices are formulated w.r.t. the tool frame $T$ at the end-effector. The full symbolic expression for the determinant of the $\mathbb{R}^{6\times6}$ matrix results in the trigonometric polynomial

$$
\begin{aligned}
\mu(\boldsymbol{q})^2 := 2r_{SE}{}^2\,r_{EW}{}^2\,\left(\mathrm{c}\,(q_4)^2 - 1\right) \Bigg( \\
+ r_{SE}{}^2\,\mathrm{c}\,(q_5)^2\,\mathrm{c}\,(q_6)^2\,\left(\mathrm{c}\,(q_2)^2 + \mathrm{c}\,(q_4)^2 - \mathrm{c}\,(q_2)^2\,\mathrm{c}\,(q_4)^2 - 1\right) \\
+ r_{EW}{}^2\,\mathrm{c}\,(q_2)^2\,\mathrm{c}\,(q_3)^2\,\left(\mathrm{c}\,(q_4)^2 + \mathrm{c}\,(q_6)^2 - \mathrm{c}\,(q_4)^2\,\mathrm{c}\,(q_6)^2 - 1\right) \\
+ \left(r_{SE}{}^2 + 2\,r_{SE}\,r_{EW}\,\mathrm{c}\,(q_4)\right)\left(\mathrm{c}\,(q_2)^2 + \mathrm{c}\,(q_6)^2 - \mathrm{c}\,(q_2)^2\mathrm{c}\,(q_6)^2 - 1\right) \\
+ \left(r_{SE}{}^2\,\mathrm{s}\,(q_4)\,\mathrm{s}\,(q_6)\,\mathrm{c}\,(q_4)\,\mathrm{c}\,(q_5)\,\mathrm{c}\,(q_6) + r_{SE}\,r_{EW}\,\mathrm{s}\,(q_4)\,\mathrm{s}\,(q_6)\,\mathrm{c}\,(q_5)\,\mathrm{c}\,(q_6)\right)\left(1 - \mathrm{c}\,(q_2)^2\right) \\
+ \Bigg(\left(r_{SE}\,r_{EW} + r_{EW}{}^2\,\mathrm{c}\,(q_4)\right)\mathrm{s}\,(q_2)\,\mathrm{s}\,(q_4)\,\mathrm{c}\,(q_2)\,\mathrm{c}\,(q_3) + r_{EW}{}^2\,\mathrm{c}\,(q_2)^2 - r_{EW}{}^2\Bigg)\left(1 - \mathrm{c}\,(q_6)^2\right) \\
\Bigg).
\end{aligned}
\tag{A1}
$$

Note that the manipulability measure $\mu$ does not depend on joints $q_1$ nor $q_7$. Further, the link lengths $\boldsymbol{r}_{BS}$ and $\boldsymbol{r}_{WT}$ do not affect manipulability.

## Appendix B. Inverse Kinematic Functions from (43)

$$q_1(\boldsymbol{p}, \lambda) := \gamma_{\text{ref}} + \text{atan2}\left(-\,\text{s}\,(\lambda)\,\text{s}\,(\theta_S)\,,\,\text{s}\,(\beta_{\text{ref}})\,\text{c}\,(\theta_S) - \text{c}\,(\beta_{\text{ref}})\,\text{c}\,(\lambda)\,\text{s}\,(\theta_S)\right) \tag{A2}$$

$$q_2(\boldsymbol{p}, \lambda) := \text{acos}\left(\text{c}\,(\beta_{\text{ref}})\,\text{c}\,(\theta_S) + \text{c}\,(\lambda)\,\text{s}\,(\beta_{\text{ref}})\,\text{s}\,(\theta_S)\right) \tag{A3}$$

$$q_3(\boldsymbol{p}, \lambda) := \text{atan2}\left(\text{s}\,(\beta_{\text{ref}})\,\text{s}\,(\lambda)\,,\,\text{c}\,(\lambda)\,\text{s}\,(\beta_{\text{ref}})\,\text{c}\,(\theta_S) - \text{c}\,(\beta_{\text{ref}})\,\text{s}\,(\theta_S)\right) \tag{A4}$$

$$q_4(\boldsymbol{p}, \lambda) := \pi - \text{acos}\left(\frac{\frac{r_{EW}^2}{2} + \frac{r_{SE}^2}{2} - \frac{r_{\text{ref}}^2}{2}}{r_{EW}\, r_{SE}}\right) \tag{A5}$$

$$q_5(\boldsymbol{p}, \lambda) := \text{atan2}\left(\text{s}\,(\gamma_{\text{EE}} - \lambda)\,\text{s}\,(\beta_{\text{EE}})\,,\,\text{s}\,(\beta_{\text{EE}})\,\text{c}\,(\theta_W)\,\text{c}\,(\gamma_{\text{EE}} - \lambda) - \text{c}\,(\beta_{\text{EE}})\,\text{s}\,(\theta_W)\right) \tag{A6}$$

$$q_6(\boldsymbol{p}, \lambda) := \text{acos}\left(\text{c}\,(\beta_{\text{EE}})\,\text{c}\,(\theta_W) + \text{s}\,(\beta_{\text{EE}})\,\text{s}\,(\theta_W)\,\text{c}\,(\gamma_{\text{EE}} - \lambda)\right) \tag{A7}$$

$$q_7(\boldsymbol{p}, \lambda) := \psi_{\text{EE}} + \text{atan2}\left(-\,\text{s}\,(\gamma_{\text{EE}} - \lambda)\,\text{s}\,(\theta_W)\,,\,\text{s}\,(\beta_{\text{EE}})\,\text{c}\,(\theta_W) - \text{c}\,(\beta_{\text{EE}})\,\text{s}\,(\theta_W)\,\text{c}\,(\gamma_{\text{EE}} - \lambda)\right) \tag{A8}$$

## Appendix C. Absolute Valued Inverse Kinematics Functions from (55a)

$$|q_1(\boldsymbol{p}, \lambda)| := \gamma_{\text{ref}} + \text{acos}\left(\frac{\text{s}\,(\beta_{\text{ref}})\,\text{c}\,(\theta_S) - \text{c}\,(\beta_{\text{ref}})\,\text{c}\,(\lambda)\,\text{s}\,(\theta_S)}{\sqrt{1 - (\text{c}\,(\beta_{\text{ref}})\,\text{c}\,(\theta_S) + \text{c}\,(\lambda)\,\text{s}\,(\beta_{\text{ref}})\,\text{s}\,(\theta_S))^2}}\right) \tag{A9}$$

$$|q_2(\boldsymbol{p}, \lambda)| := \text{acos}\left(\text{c}\,(\beta_{\text{ref}})\,\text{c}\,(\theta_S) + \text{c}\,(\lambda)\,\text{s}\,(\beta_{\text{ref}})\,\text{s}\,(\theta_S)\right) \tag{A10}$$

$$|q_3(\boldsymbol{p}, \lambda)| := \pi - \text{acos}\left(\frac{\text{c}\,(\beta_{\text{ref}})\,\text{s}\,(\theta_S) - \text{c}\,(\lambda)\,\text{s}\,(\beta_{\text{ref}})\,\text{c}\,(\theta_S)}{\sqrt{1 - (\text{c}\,(\beta_{\text{ref}})\,\text{c}\,(\theta_S) + \text{c}\,(\lambda)\,\text{s}\,(\beta_{\text{ref}})\,\text{s}\,(\theta_S))^2}}\right) \tag{A11}$$

$$|q_4(\boldsymbol{p}, \lambda)| := \pi - \text{acos}\left(\frac{\frac{r_{EW}^2}{2} + \frac{r_{SE}^2}{2} - \frac{r_{\text{ref}}^2}{2}}{r_{EW}\, r_{SE}}\right) \tag{A12}$$

$$|q_5(\boldsymbol{p}, \lambda)| := \pi - \text{acos}\left(\frac{\text{c}\,(\beta_{\text{EE}})\,\text{s}\,(\theta_W) - \text{s}\,(\beta_{\text{EE}})\,\text{c}\,(\theta_W)\,\text{c}\,(\gamma_{\text{EE}} - \lambda)}{\sqrt{1 - (\text{c}\,(\beta_{\text{EE}})\,\text{c}\,(\theta_W) + \text{s}\,(\beta_{\text{EE}})\,\text{s}\,(\theta_W)\,\text{c}\,(\gamma_{\text{EE}} - \lambda))^2}}\right) \tag{A13}$$

$$|q_6(\boldsymbol{p}, \lambda)| := \text{acos}\left(\text{c}\,(\beta_{\text{EE}})\,\text{c}\,(\theta_W) + \text{s}\,(\beta_{\text{EE}})\,\text{s}\,(\theta_W)\,\text{c}\,(\gamma_{\text{EE}} - \lambda)\right) \tag{A14}$$

$$|q_7(\boldsymbol{p}, \lambda)| := \psi_{\text{EE}} + \text{acos}\left(\frac{\text{s}\,(\beta_{\text{EE}})\,\text{c}\,(\theta_W) - \text{c}\,(\beta_{\text{EE}})\,\text{s}\,(\theta_W)\,\text{c}\,(\gamma_{\text{EE}} - \text{la})}{\sqrt{1 - (\text{c}\,(\beta_{\text{EE}})\,\text{c}\,(\theta_W) + \text{s}\,(\beta_{\text{EE}})\,\text{s}\,(\theta_W)\,\text{c}\,(\gamma_{\text{EE}} - \text{la}))^2}}\right) \tag{A15}$$

**Appendix D. Admissible Null Space Parameter Functions from** (56)

$$\lambda_1^{\lim}(\theta_S, \gamma_{\text{ref}}, \beta_{\text{ref}}, q^{\max}) :=$$

$$\left\{ \begin{array}{l} \pm\left(\pi - \text{acos}\left(\frac{\sqrt{s(\theta_S)^2 - s(\gamma_{\text{ref}} - q^{\max})^2 s(\beta_{\text{ref}})^2} + c(\beta_{\text{ref}})\,s(\beta_{\text{ref}})\,c(\theta_S)\,s(\theta_S)\left(1 - c(\gamma_{\text{ref}} - q^{\max})^2\right)}{|s(\theta_S)|\,|c(\gamma_{\text{ref}} - q^{\max})|\,s(\theta_S)^2\left(s(\gamma_{\text{ref}} - q^{\max})^2 s(\beta_{\text{ref}})^2 - 1\right)}\right)\right) \\[3mm] \pm\left(\text{acos}\left(\frac{\sqrt{s(\theta_S)^2 - s(\gamma_{\text{ref}} - q^{\max})^2 s(\beta_{\text{ref}})^2} + c(\beta_{\text{ref}})\,s(\beta_{\text{ref}})\,c(\theta_S)\,s(\theta_S)\left(c(\gamma_{\text{ref}} - q^{\max})^2 - 1\right)}{|s(\theta_S)|\,|c(\gamma_{\text{ref}} - q^{\max})|\,s(\theta_S)^2\left(s(\gamma_{\text{ref}} - q^{\max})^2 s(\beta_{\text{ref}})^2 - 1\right)}\right)\right) \end{array} \right\} \quad \text{(A16)}$$

$$\lambda_2^{\lim}(\theta_S, \beta_{\text{ref}}, q^{\max}) := \left\{ \pm\,\text{acos}\left(\frac{c(q^{\max}) - c(\beta_{\text{ref}})\,c(\theta_S)}{s(\beta_{\text{ref}})\,s(\theta_S)}\right) \right\} \quad \text{(A17)}$$

$$\lambda_3^{\lim}(\theta_S, \beta_{\text{ref}}, q^{\max}) :=$$

$$\left\{ \begin{array}{l} \pm\left(\pi - \text{acos}\left(\frac{\sqrt{-c(\beta_{\text{ref}})^2 - c(q^{\max})^2 c(\theta_S)^2 + c(q^{\max})^2 + c(\theta_S)^2} + c(\beta_{\text{ref}})\,c(\theta_S)\,s(\theta_S)\left(1 - c(q^{\max})^2\right)}{|c(q^{\max})|\,s(\beta_{\text{ref}})\left(s(q^{\max})^2 s(\theta_S)^2 - 1\right)}\right)\right) \\[3mm] \pm\left(\text{acos}\left(\frac{\sqrt{-c(\beta_{\text{ref}})^2 - c(q^{\max})^2 c(\theta_S)^2 + c(q^{\max})^2 + c(\theta_S)^2} + c(\beta_{\text{ref}})\,c(\theta_S)\,s(\theta_S)\left(c(q^{\max})^2 - 1\right)}{|c(q^{\max})|\,s(\beta_{\text{ref}})\left(s(q^{\max})^2 s(\theta_S)^2 - 1\right)}\right)\right) \end{array} \right\} \quad \text{(A18)}$$

$$\lambda_5^{\lim}(\theta_W, \beta_{\text{EE}}, q^{\max}) := \gamma_{\text{EE}} - \lambda_3^{\lim}(\theta_W, \beta_{\text{EE}}, q^{\max}) \quad \text{(A19)}$$

$$\lambda_6^{\lim}(\theta_W, \beta_{\text{EE}}, q^{\max}) := \gamma_{\text{EE}} - \lambda_2^{\lim}(\theta_W, \beta_{\text{EE}}, q^{\max}) \quad \text{(A20)}$$

$$\lambda_7^{\lim}(\theta_W, \gamma_{\text{EE}}, \beta_{\text{EE}}, q^{\max}) := \gamma_{\text{EE}} - \lambda_1^{\lim}(\theta_W, \gamma_{\text{EE}}, \beta_{\text{EE}}, q^{\max}). \quad \text{(A21)}$$

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
