# Peer review of "Efficient Closed-Form Task Space Manipulability for a 7-DOF Serial Robot"

_robotics, doi:10.3390/robotics8040098_

Round 1
Reviewer 1 Report
Review of:
"Efficient Closed-Form Task Space Manipulability for a 7-DOF Serial Robot"
In this paper, the authors derive analytical expressions for a conventional 7-DOF serial robot structure, that enable direct 6 evaluation of manipulability from a reduced task space parametrizations.
In general terms, the paper is well written and well presented, however, some minor issues need to be improved. Some comments are presented next.
1 - Although most of the images are in a vectorized format, some of them are not, difficulting the visualization a compromising the quality in the printed version. I suggest re-size all images and font sizes in them considering that a high number of readers like to read in printed format. Also, I suggest improving some details in the chosen colors in figures once that several details are only visible in the electronic format and not in a BW printed format.
2 - In page 8, line 212 is written "We consequently choose to formulate the Jacobi matrix...", it must be written, "We consequently choose to formulate the Jacobian matrix ".
3 - In page 14, line 258 is written "All closed-form expression resulting from the IK mapping are fully listed in the appendix....", it must be clarify in which appendix are listed (You have four appendices).
Author Response
Dear reviewer,
we thank you very much for your review on our manuscript “Efficient Closed-Form Task Space Manipulability for a 7-DOF Serial Robot”. Please find attached a comprehensive version of the “Reviewer(s)' Comments to Author” file we received, including in-line discussions.
For better readability, we state all our our comments in orange. All changes in the revised manuscript are accordingly marked in orange font colour too. Further, we numbered all your remarks and accordingly tagged the phrases in the revised manuscript at the right margin. Please find at the end of the manuscript additionally a complete list of all tags.
We look forward to hearing from you in due time regarding our submission and to respond to any further questions and comments you may have.
Kind regards,
Gerold Huber and Dirk Wollherr

Reviewer 2 Report
The authors of “Efficient Closed-Form Task Space Manipulability for a 7-DoF Serial Robot” present a framework for parametrization of task and null spaces which leads to a new inverse kinematic solver and an admissible parameter mapping. This framework is focused on manipulation task for 7-DoF Serial Robot, with a mathematical formulation done for robots of generic number of DoF. Because the current interest in robotic manipulation task nowadays inside the robotics community and industry, works for resolving efficiently task space manipulability problems are welcomed. But, this work presents some doubts which must be clarify.
Although the authors tried to justify and argument the fact that a 6DoF robot results in slightly better probability density function than for the discussed 7DoF mechanism (Section 4.3). Because this is really interesting (goes in opposite direction to the rest of the community), the discussion of this fact should be improved.
Since this work is based the manipulability in the well-known Yoshikawa’s work. Many other approaches presenting different inverse kinematic solvers for manipulation are presented in the literature. The author must compare its purpose against other common approaches.
I’ve detected some typos inside the text, the manuscript must be proof reading after the publication.
Some of the typos and text style errors found are the following:
* page 1 “… a parametrization of SE(3) rather then the joint.” -- “rather then” should be “rather than”
* page 1 “… of the Inverse Kinematic (IK) with the defined capability measure.” -- capability of measurement.
* page 4 after line 163 “Given is a n-DOF serial robot with the forward kinematics” -- unnecessary “is”
* page 8 line 211 “the first joint rotates the the whole kinematics” -- duplicate the the
* page 24 “For a minimal working example, we us direct interpolation” -- change us by use
* page 24 line 391 “Given are For two random poses as depicted in Figure 11 to the left” -- it seems that “are For” is an error.
* page 25. Author Contribution paragraph is inside conclusions paragraph.
Author Response
Dear reviewers,
we thank you very much for your review of our manuscript “Efficient Closed-Form Task Space Manipulability for a 7-DOF Serial Robot”. Please find attached a comprehensive version of the “Reviewer(s)' Comments to Author” file we received, including in-line discussions.
For better readability, we state all our comments in orange. All changes in the revised manuscript are accordingly marked in orange font colour too. Further, we numbered all your remarks and accordingly tagged the phrases in the revised manuscript at the right margin. Please find at the end of the manuscript additionally a complete list of all tags.
We look forward to hearing from you in due time regarding our submission and to respond to any further questions and comments you may have.
Kind regards,
Gerold Huber and Dirk Wollherr

Reviewer 3 Report
This paper presents a new state and null-space parametrization to evaluate task space manipulability. However, there are some issues to be pointed out:
Could the authors explain if the approach has been tested in a real manipulation task? The conclusions section is poor
Author Response

(The authors gave the same response as above.)

Round 2
Reviewer 2 Report
I appreciate that authors considered valuable my suggestions and that they carefully followed all of them. I have not more comments to add.